# Polo-like kinase 1 induces epithelial-to-mesenchymal transition and promotes epithelial cell motility by activating CRAF/ERK signaling

**Jianguo Wu[1,2,3], Andrei I Ivanov[1,2,3], Paul B Fisher[1,2,3], Zheng Fu[1,2,3]\***

[1]Department of Human and Molecular Genetics, Virginia Commonwealth University School of Medicine, Richmond, United States; [2]VCU Institute of Molecular Medicine, Virginia Commonwealth University School of Medicine, Richmond, United States; [3]VCU Massey Cancer Center, Virginia Commonwealth University School of Medicine, Richmond, United States

**Abstract** Polo-like kinase 1 (PLK1) is a key cell cycle regulator implicated in the development of various cancers, including prostate cancer. However, the functions of PLK1 beyond cell cycle regulation remain poorly characterized. Here, we report that *PLK1* overexpression in prostate epithelial cells triggers oncogenic transformation. It also results in dramatic transcriptional reprogramming of the cells, leading to epithelial-to-mesenchymal transition (EMT) and stimulation of cell migration and invasion. Consistently, PLK1 downregulation in metastatic prostate cancer cells enhances epithelial characteristics and inhibits cell motility. The signaling mechanisms underlying the observed cellular effects of PLK1 involve direct PLK1-dependent phosphorylation of CRAF with subsequent stimulation of the MEK1/2-ERK1/2-Fra1-ZEB1/2 signaling pathway. Our findings highlight novel non-canonical functions of PLK1 as a key regulator of EMT and cell motility in normal prostate epithelium and prostate cancer. This study also uncovers a previously unanticipated role of PLK1 as a potent activator of MAPK signaling.

**\*For correspondence:** zheng.fu@vcuhealth.org

## Introduction

Mammalian polo-like kinase 1 (PLK1) is a serine/threonine kinase that plays key roles in the regulation of the cell cycle (*Barr et al., 2004*; *Llamazares et al., 1991*). It contains a conserved N-terminal kinase catalytic domain and a C-terminal polo-box domain (PBD) that is involved in substrate binding. PLK1 mediates almost every stage of cell division, including mitotic entry, centrosome maturation, bipolar spindle formation, sister chromatid segregation, mitotic exit, and cytokinesis execution (*Barr et al., 2004*). In addition to its canonical role in mitosis and cytokinesis, recent studies suggest that PLK1 may have other important functions such as regulation of microtubule dynamics, DNA replication, chromosome dynamics, p53 activity, and recovery from the G2 DNA damage checkpoint (*Liu et al., 2010*; *Song et al., 2011*).

*PLK1* is overexpressed in a variety of human tumors and its expression level often correlates with increased cellular proliferation, enhanced metastatic potential, and poor prognosis in cancer patients (*Cholewa et al., 2013*; *Takai et al., 2005*). *PLK1* is frequently (>50%) overexpressed in prostate cancer (PCa), and *PLK1* overexpression is linked to higher tumor grade (*Weichert et al., 2004*), suggesting that PLK1 may play a pivotal role in PCa etiology. Constitutive expression of *Plk1* in NIH/3T3 cells causes oncogenic foci formation and these transformed cells are tumorigenic in nude mice (*Smith et al., 1997*). In contrast, depleting PLK1 in U2OS cells abrogates anchorage-independent

**eLife digest** Living cells grow and divide via a series of events called the cell cycle. If this process is disturbed in animals, it can lead to cancer. In the later stages of tumor development, cancer cells frequently change their structure and behavior in a process called the epithelial-to-mesenchymal transition (EMT), which enables them to migrate and form new tumors around the body.

A protein called Polo-like kinase 1 (PLK1) plays a central role in the cell cycle and has been implicated in the development of various cancers, including prostate cancer. Recent evidence suggests that PLK1 also has other roles in cells, but it is not clear how much they contribute to the development of cancer.

Wu et al. studied PLK1 in human cells and mice and showed that manipulating healthy prostate epithelial cells to produce more PLK1 caused the cells to go through the EMT and increased their ability to migrate. In other experiments, the levels of PLK1 in prostate cancer cells were deliberately lowered, which caused the cells to change to become more like epithelial cells and become less mobile. Wu et al. also investigated how PLK1 promotes the EMT and cell migration. These experiments showed that PLK1 activates a protein that controls an important chain of signaling events called the ERK/MAPK pathway, which is essential for cell growth and migration.

Wu et al.'s findings uncover a new role for PLK1 in promoting the spread of cancer cells around the body. A future challenge is to find out how PLK1 is regulated in people with prostate cancer and whether the EMT is involved in promoting other processes in cancer cells.

growth (*Eckerdt et al., 2005*). These results highlight PLK1 as a possible driver of oncogenic transformation, although it remains unclear if PLK1 itself is sufficient to induce tumor development. It has been suggested that PLK1 controls cancer development through multiple mechanisms that include canonical regulation of mitosis and cytokinesis, as well as modulation of DNA replication and cell survival (*Deeraksa et al., 2013*; *Luo and Liu, 2012*). Importantly, previous studies reported that increased PLK1 expression levels positively correlate with the invasiveness of colorectal, breast, and thyroid tumors (*Han et al., 2012*; *Rizki et al., 2007*; *Zhang et al., 2012*). These data imply a possible role for PLK1 in tumor invasion and metastasis; however, direct evidence supporting this hypothesis and mechanisms of the proinvasive activity of PLK1 during PCa progression are lacking.

In this study, we investigated the roles of PLK1 in regulating the motility of prostate epithelial cells and PCa cells. Our data highlight PLK1 as a crucial positive regulator of different modes of cell migration. This pro-migratory activity of PLK is mediated by induction of the epithelial-to-mesenchymal transition (EMT) via activation of the CRAF/MEK/ERK/Fra1/ZEB1/2 signaling cascade.

## Results

### *PLK1* overexpression induces prostate epithelial cell transformation and stimulates cell motility

It has been reported that PLK1 is frequently overexpressed in human PCa (*Weichert et al., 2004*). To examine the expression level and activity status of PLK1 in a panel of PCa cell lines, we performed immunoblotting analysis using antibodies that recognize total PLK1 or its active form, phosphorylated at Tyrosine 210 (pT210). Both the protein abundance and activity of PLK1 were elevated in PCa cell lines when compared to RWPE-1 cells (immortalized normal prostate epithelial cells; *Figure 1A*), which is consistent with the PLK1 expression profile in PCa tissue specimens reported by another group (*Weichert et al., 2004*). Moreover, PLK1 was differentially expressed and/or activated in PCa cells (higher in the metastatic PCa cell lines [DU145, C4-2B and PC3] and lower in the non-metastatic cell lines [LNCaP and LAPC4]; *Figure 1A*).

In order to define the potential oncogenic role of *PLK1* upregulation in PCa, we overexpressed PLK1 in RWPE-1 cells. RWPE-1 cells are derived from normal human prostate epithelial cells and are immortalized with human papillomavirus 18 E7 proteins (*Bello et al., 1997*). In contrast to E6-immortalized cells, RWPE-1 cells express p53 and have a functional p53-dependent checkpoint

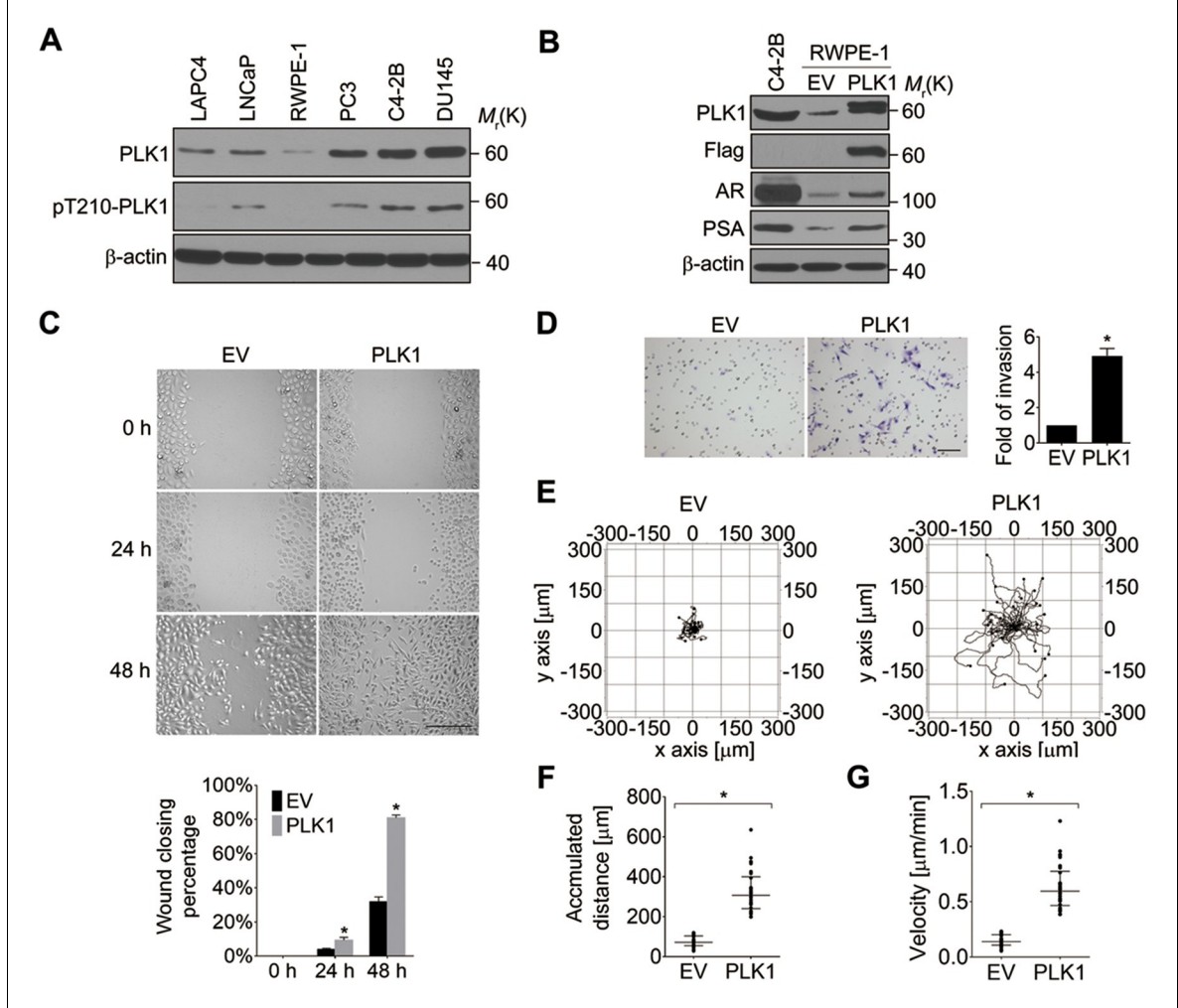

**Figure 1.** Ectopic expression of PLK1 in RWPE-1 cells promotes cell motility. (A) Cell lysates were prepared from the indicated PCa cell lines and subjected to Western blots in order to detect the level and activity of PLK1 protein using anti−PLK1 and anti−PLK1(pT210) antibodies, respectively. β-actin was used as a loading control. (B) RWPE-1 cells were infected with lentivirus encoding Flag-PLK1 (PLK1) or empty vector (EV). The protein levels of PLK1, AR, PSA, and β-actin were determined by Western blot. C4-2B cells with high endogenous PLK1 expression were included for comparison. (C) Control RWPE-1 and RWPE-1–PLK1 cells were subjected to a wound healing assay. The figure shows representative images as well as calculated percentage of wound closure during 48 hr of cell migration. Scale bar, 500 μm. (D) *In vitro* Matrigel invasion assay. The figure shows representative images of invaded cells and quantification of the relative number of cells that invaded over 48 hr. The data are presented as the mean ± s.e.m. *p<0.01, two-tailed Student's *t*-test. Scale bar, 100 μm. (E–G) Time-lapse video microscopy motility experiments to monitor random migration of control and PLK1-overexpressing RWPE-1 cells. The trajectories of individual cells of different experimental groups (E), track distance (F), and velocity of cell migration (G) are shown. Horizontal bars in F-G represent median and interquartile range. Each dot represents a single-cell measurement. Thirty cells per experimental group were measured. *p<0.01, two-tailed Mann-Whitney rank sum tests.

The following figure supplements are available for figure 1:

**Figure supplement 1.** Androgen receptor (*AR*) mRNA is expressed in RWPE-1 cells.

**Figure supplement 2.** Ectopic expression of PLK1 in RWPE-1 cells induces cellular transformation and tumorigenicity.

**Figure supplement 3.** Ectopic expression of PLK1 in PrEC cells promotes cell migration and invasion.

(*Bello et al., 1997*; *Roh et al., 2008*). Furthermore, they express luminal cytokeratins and do not grow in soft agar or form tumors in nude mice. We detected low levels of androgen receptor (AR) expression in RWPE-1 cells by quantitative real-time RT-PCR and immunoblotting (*Figure 1—figure*

*supplement 1*, *Figure 1B*). Therefore, RWPE-1 cells provide an excellent model for studying normal prostate epithelial functions, prostate epithelial transformation, and different stages of prostate carcinogenesis (*Bello et al., 1997*; *Roh et al., 2008*).

RWPE-1 cells were infected with lentivirus containing human *PLK1* cDNA. To avoid clonal variation, stable cell lines were established from a mixed population of multiple clones. Cells transduced with empty lentivirus served as a control. RWPE-1–PLK1 cells were designed to express PLK1 at a level comparable to asynchronous metastatic C4-2B PCa cells (*Figure 1B*) in order to determine the effect of cell cycle-independent *PLK1* overexpression on PCa development. We examined the ability of PLK1-overexpressing cells to grow in soft agar, a property that frequently correlates with cell tumorigenicity. Under these conditions, control RWPE-1 cells formed few, if any, colonies in soft agar, whereas RWPE-1–PLK1 cells showed robust colony formation after 3 weeks of growth (*Figure 1—figure supplement 2A*). This indicates that *PLK1* overexpression is sufficient for cellular transformation of RWPE-1 cells. To assess *in vivo* tumorigenicity of PLK1-overexpressing prostate epithelial cells, control RWPE-1 or RWPE-1–PLK1 cells were injected subcutaneously into the flanks of NOD/SCID/$\gamma_c^{null}$ (NSG) mice (*Fu et al., 2003*). Six weeks after implantation, 100% (7 out of 7) of mice injected with RWPE-1–PLK1 cells developed primary tumors with an average size of 2,356 ± 589 (SEM) mm (*Barr et al., 2004*) (*Figure 1—figure supplement 2B*). In contrast, no tumors developed in mice injected with control RWPE-1 cells (*Figure 1—figure supplement 2B*). Interestingly, lung micrometastases were found in 5 of the 7 mice injected with RWPE-1–PLK1 cells. The micrometastases stained positive for PSA (*Figure 1—figure supplement 2C*), indicating that they were of human origin and were derived from RWPE-1–PLK1 cells (*Figure 1B*) (*Bello et al., 1997*). This data suggests that *PLK1* overexpression not only leads to oncogenic transformation of prostate epithelial cells, but may also drive PCa metastasis.

Based on the results of our tumor xenograft experiments implicating PLK1 in PCa metastasis, we next investigated the role of PLK1 in regulating prostate epithelial cell motility *in vitro*. Two classical assays of cell motility were used: one examined the collective cell migration during closure of planar epithelial wounds and the other analyzed the invasion of individual cells into Matrigel. *Figure 1C,D* demonstrates that *PLK1* overexpression significantly accelerated both wound closure and Matrigel invasion of RWPE-1 cells. Importantly, *PLK1* overexpression also promoted migration and invasion of PrEC (human primary prostate epithelial cells) (*Rudolph et al., 2009*) (*Figure 1—figure supplement 3*), thereby indicating that this pro-migratory effect is not a peculiar feature of the RWPE-1 cell line. Since PLK1 is a crucial regulator of the cell cycle, it may promote epithelial cell motility indirectly by stimulating cell proliferation. To rule out this indirect effect, we compared the random movement of individual RWPE-1–PLK1 cells to that of control RWPE-1 cells using time-lapse microscopy. Both migratory distance and velocity were significantly higher in RWPE-1-PLK1 cells compared to control

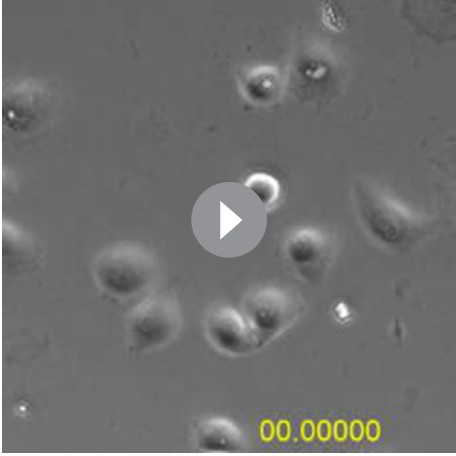

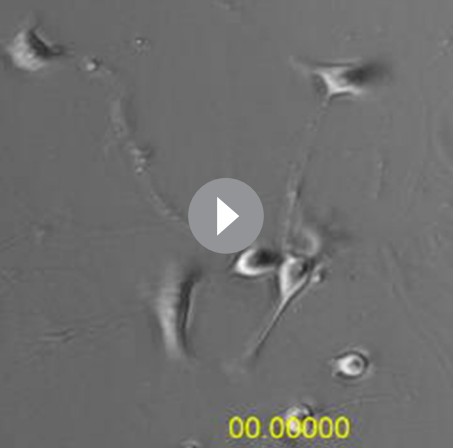

**Video 1.** Video showing the random migration of control RWPE-1 cells.

**Video 2.** Video showing the random migration of RWPE-1–PLK1 cells.

RWPE-1 cells (*Figure 1E–G*, *Videos 1*, *2*). These data indicate that *PLK1* overexpression stimulates cell motility in a proliferation-independent fashion.

## *PLK1* overexpression induces EMT in prostate epithelial cells

*PLK1* overexpression in RWPE-1 cells causes the cells to change shape from an orthogonal epithelial cell morphology to a spindle-shaped fibroblast-like morphology (*Figure 2A*), reminiscent of cells having undergone EMT. EMT is believed to play key roles in tumor progression and metastasis by

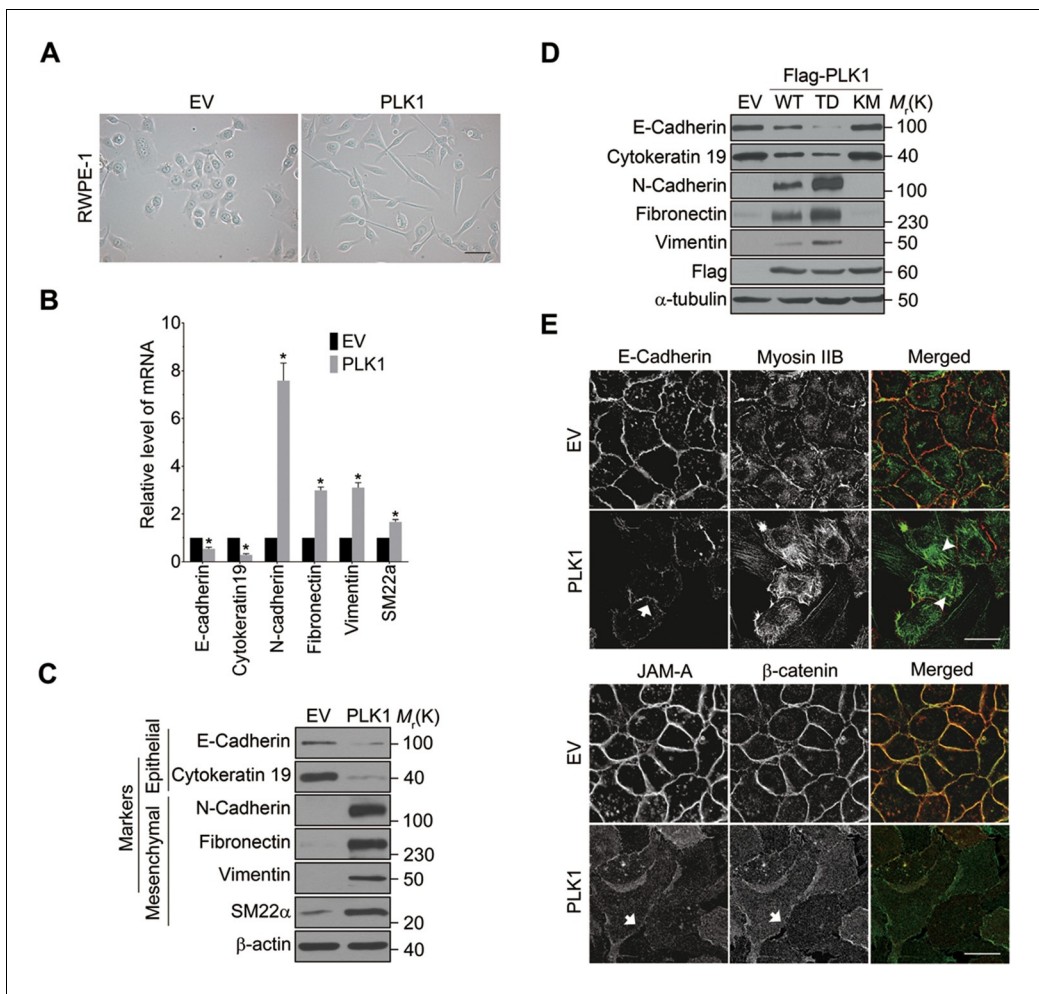

**Figure 2.** Overexpressing PLK1 in RWPE-1 cells induces EMT. (**A**) Representative phase-contrast images of control RWPE-1 and RWPE-1–PLK1 cells. Scale bar = 50 μm. (**B**, **C**) mRNA and protein expression of different EMT markers in RWPE-1–PLK1 cells (PLK1) and vector control cells (EV) were examined by real-time RT-PCR (**B**) and Western blot (**C**), respectively. The data are presented as the mean ± s.e.m. *p<0.01, two-tailed Student's *t*-test. (**D**) RWPE-1 cells were infected with lentivirus expressing wild-type PLK1 (WT), constitutively active T210D (TD), or kinase-dead K82M (KM) PLK1 mutants. Expression of EMT markers was examined by immunoblotting. (**E**) The architecture of adherens junctions (E-cadherin and β-catenin labeling), tight junctions (JAM-A), and the actomyosin cytoskeleton (myosin IIB) in control and PLK1-overexpressing cells was evaluated by immunolabeling and confocal microscopy. Arrows indicate disrupted adherens junctions and tight junctions, whereas arrowheads point to basal stress fibers in PLK1-overexpressing cells. Scale bar, 20 μm.

The following figure supplements are available for figure 2:

**Figure supplement 1.** The effect of PLK1 overexpression on EMT is independent of its cell cycle function.

**Figure supplement 2.** Overexpression of PLK1 in PrEC cells induces EMT.

enabling a switch from the stationary epithelial-like cell phenotype to the motile mesenchymal phenotype. On a molecular level, EMT is characterized by the decreased expression of epithelial markers and the increased expression of mesenchymal markers. In order to investigate whether *PLK1* overexpression triggers EMT in RWPE-1 cell, we examined the expression of most of the characteristic epithelial and mesenchymal markers. Remarkably, relative to control RWPE-1 cells, RWPE-1–PLK1 cells downregulated epithelial markers (E-cadherin and cytokeratin 19) and upregulated mesenchymal markers (N-cadherin, vimentin, fibronectin, and SM22), at both the mRNA and protein levels (*Figure 2B,C*). The switch from epithelial to mesenchymal markers did not depend on the stages of the cell cycle (*Figure 2—figure supplement 1*). In agreement with the results obtained in RWPE-1 cells, *PLK1* overexpression also induced an EMT-like phenotype in PrEC cells (*Figure 2—figure supplement 2*). Furthermore, we compared induction of EMT in cells expressing wild-type (WT), constitutively active (TD), or kinase-defective (KM) PLK1. Constitutively active PLK1 induced the most robust reprogramming of gene expression in RWPE-1 cells, whereas expression of kinase inactive PLK1 failed to induce EMT (*Figure 2D*). These results suggest that a PLK1-mediated phosphorylation event contributes to the induction of EMT in prostate epithelial cells. Importantly, *PLK1* overexpression in RWPE-1 cells disrupted localization of E-cadherin, β-catenin, and junctional adhesion molecule (JAM)-A at the areas of cell-cell contacts, thereby indicating profound disassembly of adherens and tight junctions (*Figure 2E*, arrows). This was accompanied by dramatic reorganization of the actomyosin cytoskeleton manifested by redistribution of non-muscle myosin IIB from the perijunctional F-actin bundles into basal stress fibers (*Figure 2E*, arrowheads). Taken together, these data demonstrate induction of EMT in PLK1-overexpressing prostate epithelial cells, which is likely to promote a pro-motile phenotype by disrupting intercellular adhesions and creating isolated mesenchymal cells with invasive properties (*Moreno-Bueno et al., 2008*).

## Downregulation of PLK1 inhibits motility of metastatic PCa cells

Given the dramatic acceleration of cell motility in PLK1-overexpressing prostate epithelial cells, we sought to investigate whether downregulation of endogenous PLK1 could attenuate the migration of PCa cells. To exclude the effects of severe PLK1 depletion on growth inhibition and cell death, stable cell lines with partial PLK1 knockdown using a lentivirus encoding shRNA that targets the 3′-UTR of PLK1 were established using 2 known metastatic PCa cell lines (DU145 and C4-2B). Downregulation of PLK1 expression in these cells was confirmed by immunoblotting (*Figure 3A*, *Figure 3—figure supplement 2A*). As shown in *Figure 3—figure supplement 1*, the partial PLK1 knockdown did not affect cell cycle progression and did not trigger cell apoptosis. Importantly, PLK1 downregulation promoted an epithelial phenotype in PCa cells by decreasing the expression of mesenchymal markers and increasing E-cadherin expression, inducing morphological alterations to a more epithelial-like appearance, and triggering the assembly of intercellular junctions (*Figure 3A−C*). Furthermore, downregulation of PLK1 expression resulted in significant attenuation of PCa cell motility based on both wound closure and Matrigel invasion assays (*Figure 3D,E*). All the observed changes induced by PLK1 knockdown were reversed by re-expression of WT PLK1, but not by kinase-defective KM PLK1 (*Figure 3A−E*).

To exclude the possibility of off-target effects of individual shRNAs, another shRNA (shPLK1#2) targeting a different site within the 3′-UTR of PLK1 was used to verify our observations (*Figure 3—figure supplement 2*). Similarly, upon partially knocking down endogenous PLK1 with yet another shRNA in 3 metastatic PCa cell lines (PC3, C4-2B, and DU145), the effects of PLK1 on EMT induction and pro-motile phenotype were reversed (*Figure 3—figure supplement 2*). Furthermore, we also used an androgen-refractory cancer of the prostate (ARCaP) model to verify the role of PLK1 in induction of EMT. ARCaP cells were derived from the ascites fluid of an 83-year-old Caucasian man diagnosed with metastatic carcinoma of the prostate (*Zhau et al., 1996*). Epithelium-like ARCaP$_E$ cells and mesenchymal-like ARCaP$_M$ cells are sublines of ARCaP cells that were isolated by single-cell dilution cloning (*Xu et al., 2006*). Interestingly, PLK1 is differentially expressed and activated in those 2 cell lines (higher in highly metastatic ARCaP$_M$ cells and lower in less metastatic ARCaP$_E$ cells; *Figure 3—figure supplement 3*), further suggesting a role for PLK1 in PCa metastasis. Upregulation of PLK1 in ARCaP$_E$ cells led to the induction of EMT, whereas downregulation of PLK1 in ARCaP$_M$ cells promoted the reversion of EMT manifested by biochemical changes as well as altered morphology, assembly of intracellular junctions, and motility (*Figure 3—figure supplement 3*). Taken

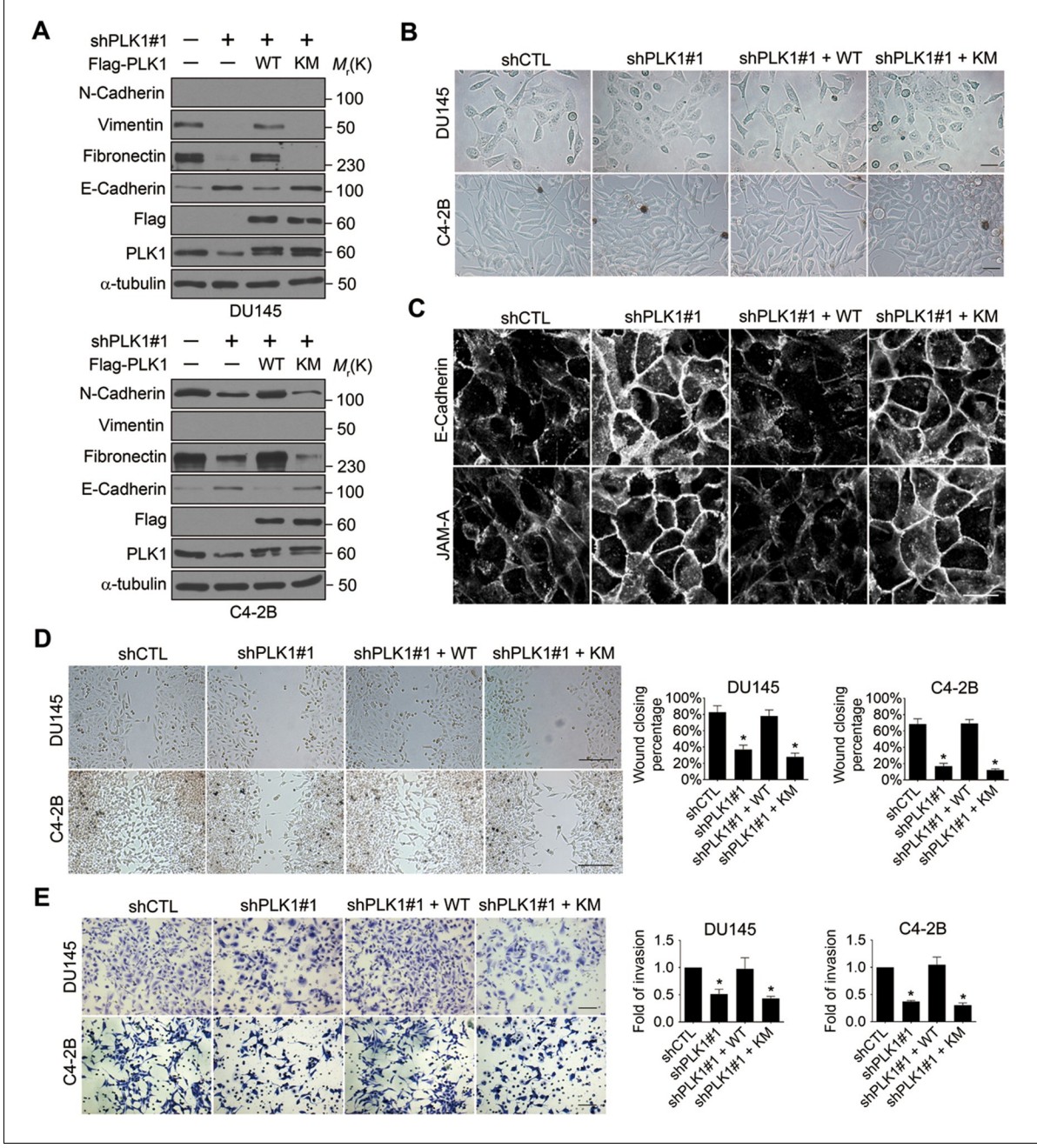

**Figure 3.** Downregulation of PLK1 reverses EMT and inhibits the motility of metastatic PCa cells. (A) Two metastatic PCa cell lines (DU145 and C4-2B) were infected with lentiviral shRNA constructs that target either the 3'-UTR of endogenous PLK1 (shPLK1#1) or serve as a control (shCTL). Wild-type (WT) or kinase-defective (KM) PLK1 were then re-expressed in PLK1 knockdown cells. The expression of PLK1 protein and EMT markers was determined by immunoblotting. In the PLK1 immunoblot, the lower bands show endogenous PLK1 expression and the upper bands show exogenous PLK1 expression. (B) Representative phase-contrast images of control, PLK1 knockdown cells, and PLK1 knockdown cells with re-expression of WT or KM PLK1. Scale bar = 50 µm. (C) The architecture of adherens junctions (E-cadherin) and tight junctions (JAM-A) in control cells and cells with PLK1 manipulation was evaluated by immunolabeling and confocal microscopy. Scale bar, 20 µm. (D, E) The effects of PLK1 knockdown on planar migration and invasion of PCa cells were measured by the wound healing assay (D) and Matrigel invasion assay (E), respectively. The data are presented as the mean ± s.e.m. *p<0.01, two-tailed Student's t-test. Scale bar, 500 µm (D), and 100 µm (E).

The following figure supplements are available for figure 3:

**Figure supplement 1.** Partial knockdown of PLK1 in PCa cells has minimal effect on cell cycle progression and cell death.

*Figure 3 continued on next page*

*Figure 3 continued*

**Figure supplement 2.** Downregulation of PLK1 results in EMT reversion and reduction of migration of metastatic PCa cells.

**Figure supplement 3.** Validation of the role of PLK1 in induction of EMT in the ARCaP cell culture model.

**Figure supplement 4.** PLK1 contributes to physiological EMT events.

together, our overexpression and knockdown experiments revealed a novel function of PLK1 as a critical regulator of prostate epithelial cell motility and EMT.

To determine whether PLK1 is involved in physiologically-relevant EMT, RWPE-1 and ARCaP$_E$ cells were treated with a combination of EMT-inducing growth factors (*Zhau et al., 2008*). Following treatment with transforming growth factor β1 (TGF-β1) and epidermal growth factor (EGF), cells underwent an EMT-like process as demonstrated by acquisition of spindle-like cell morphology and switched expression of epithelial and mesenchymal markers (*Figure 3—figure supplement 4*). Although growth-factor−induced EMT was not accompanied by increased expression or activity of PLK1, this process was partially blocked by a low dose of a potent pharmacological inhibitor of PLK1: BI2536 (*Steegmaier et al., 2007*) (*Figure 3—figure supplement 4*). The phosphorylation of FoxM1 (a PLK1 substrate) at S724 was used as a readout for PLK1 activity (*Figure 3—figure supplement 4*). These data suggest that PLK1 could be important for EMT-induced by physiological and pathophysiological stimuli.

## *PLK1* overexpression promotes EMT and motility of prostate epithelial cells by upregulating ZEB proteins

We next sought to elucidate the signaling pathways downstream of oncogenic PLK1 that are responsible for the EMT induction and increased motility of prostate epithelial cells. Expressional reprogramming of epithelial cells undergoing EMT is known to be mediated by several transcriptional regulators, most notably Snail1, Snail2 (Slug), ZEB1, ZEB2, E47, and Twist (*Sánchez-Tilló et al., 2012*). Therefore, we examined which of these transcriptional regulators are induced by *PLK1* overexpression in prostate epithelial cells. Quantitative real-time PCR analysis demonstrated marked and selective upregulation of ZEB1 and ZEB2 in RWPE-1–PLK1 cells, whereas expression of Snail1, Snail2, E47, and Twist was not significantly changed (*Figure 4A*). Furthermore, increased protein expression of ZEB1 and ZEB2 was observed by immunoblotting in PLK1-overexpressing RWPE-1 cells (*Figure 4B*) and PrEC cells (*Figure 4—figure supplement 1*).

In order to determine if ZEB1 and/or ZEB2 have a causal role in PLK1-dependent induction of EMT, we depleted these transcriptional regulators individually or in combination, in RWPE-1–PLK1 cells (*Figure 4C*). Downregulation of ZEB2 increased the expression of E-cadherin and cytokeratin 19, while it decreased the levels of N-cadherin, fibronectin, and vimentin; conversely, depletion of ZEB1 was less effective and primarily decreased vimentin expression (*Figure 4C*). Interestingly, dual knockdown of ZEB1 and ZEB2 completely reversed expressional reprogramming of EMT markers in RWPE-1–PLK1 cells (*Figure 4C*). Depletion of ZEB1 and ZEB2 proteins not only eliminated the biochemical signature of the EMT, but also reversed the functional effects of *PLK1* overexpression. For example, either individual or co-knockdown of ZEB1 and ZEB2 restored assembly of adherens and tight junctions in RWPE-1–PLK1 cells (*Figure 4D*). Moreover, downregulation of ZEB1 or ZEB2 significantly decreased planar migration and Matrigel invasion of RWPE-1–PLK1 cells (*Figure 4E,F*). Remarkably, dual knockdown of ZEB1 and ZEB2 completely abolished PLK1-induced motility of RWPE-1–PLK1 cells (*Figure 4E,F*). These findings indicate that ZEB1 and ZEB2 play causal roles in PLK1-induced EMT and motility in prostate epithelial cells.

## ERK1/2-Fra1 signaling mediates PLK1-induced cell motility and EMT

Next, we investigated the signaling pathways that are responsible for the induction of ZEB1 and ZEB2 in PLK1-overexpressing prostate epithelial cells. Aberrant activation of the mitogen-activated protein kinase/extracellular signal-regulated kinase (MAPK/ERK) pathway has been shown to contribute to tumor invasion and progression, and has recently been linked to the EMT process (*Reddy et al., 2003*; *Shin et al., 2010*). Therefore, we examined the effect of *PLK1* overexpression

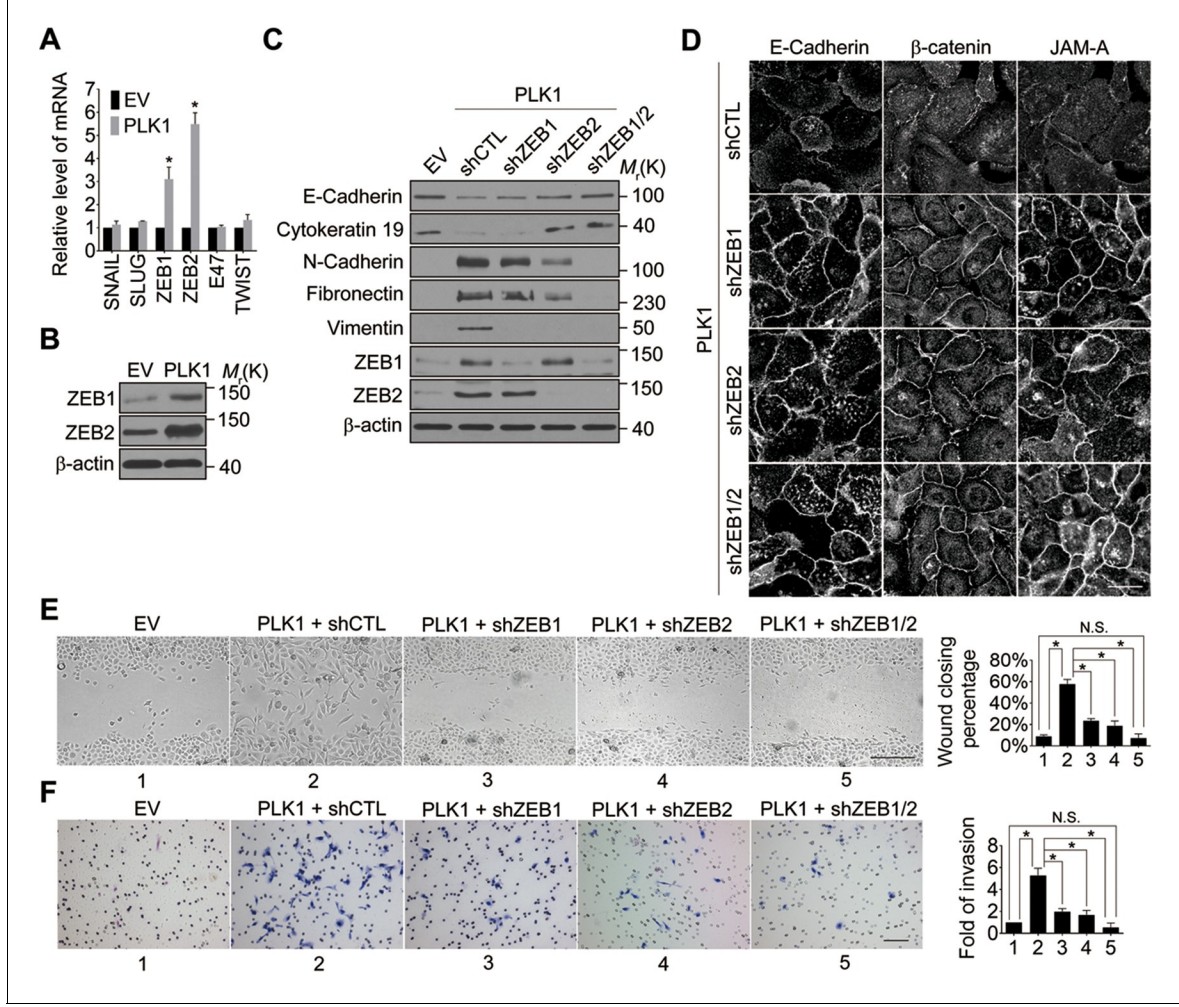

**Figure 4.** Both ZEB1 and ZEB2 play a causal role in PLK1-induced EMT and increased motility of prostate epithelial cells. (A) Expression of different EMT-inducing transcription factors was examined in control and PLK1-overexpressing RWPE-1 cells by quantitative real-time RT-PCR. mRNA expression of genes of interest was normalized by the level of *glyceraldehyde phosphate dehydrogenase* (*GAPDH*) mRNA and is presented as relative expression. The data are presented as the mean ± s.e.m. *p<0.05, two-tailed Student's *t*-test. (B) The levels of ZEB1 and ZEB2 proteins in control and PLK1-overexpressing cells were examined by immunoblotting. (C−F) The effects of either individual shRNA-mediated downregulation of ZEB1 and ZEB2, or their dual knockdown in PLK1-induced EMT (C); cell-cell junctional disassembly (D); planar cell migration (E); and cell invasion (F) were determined as described in *Figures 1–2*. N.S.: no significant difference. Scale bar, 20 μm (D), 500 μm (E), and 100 μm (F).

The following figure supplement is available for figure 4:

**Figure supplement 1.** Elevated PLK1 levels result in activation of ERK, and upregulation of ZEB1, ZEB2, and Fra1 in PrEC cells.

on the activation status of key molecular constituents of the ERK signaling cascade. *PLK1* overexpression in RWPE-1 cells resulted in increased phosphorylation of ERK1/2 and its upstream kinase MEK1/2 (*Figure 5A*), thereby indicating activation of MEK1/2-ERK1/2 signaling. Consistently, we also detected enhanced phosphorylation of ERK1/2 in PLK1-overexpressing PrEC cells (*Figure 4— figure supplement 1*). In order to determine the potential causal role of the MEK/ERK pathway in PLK1-induced EMT and accelerated motility of prostate epithelial cells, we blocked this pathway with RO5126766, a highly selective dual inhibitor of MEK and its upstream kinase, RAF (*Martinez-Garcia et al., 2012*). Exposure of RWPE-1–PLK1 cells to RO5126766 abrogated ERK activation, reduced expression of ZEB1 and ZEB2, and completely reversed the biochemical manifestation of EMT (*Figure 5B*). Moreover, inhibition of MEK/ERK signaling restored assembly of intercellular junctions (*Figure 5C*) and inhibited the increased wound closure and Matrigel invasion of RWPE-1–PLK1

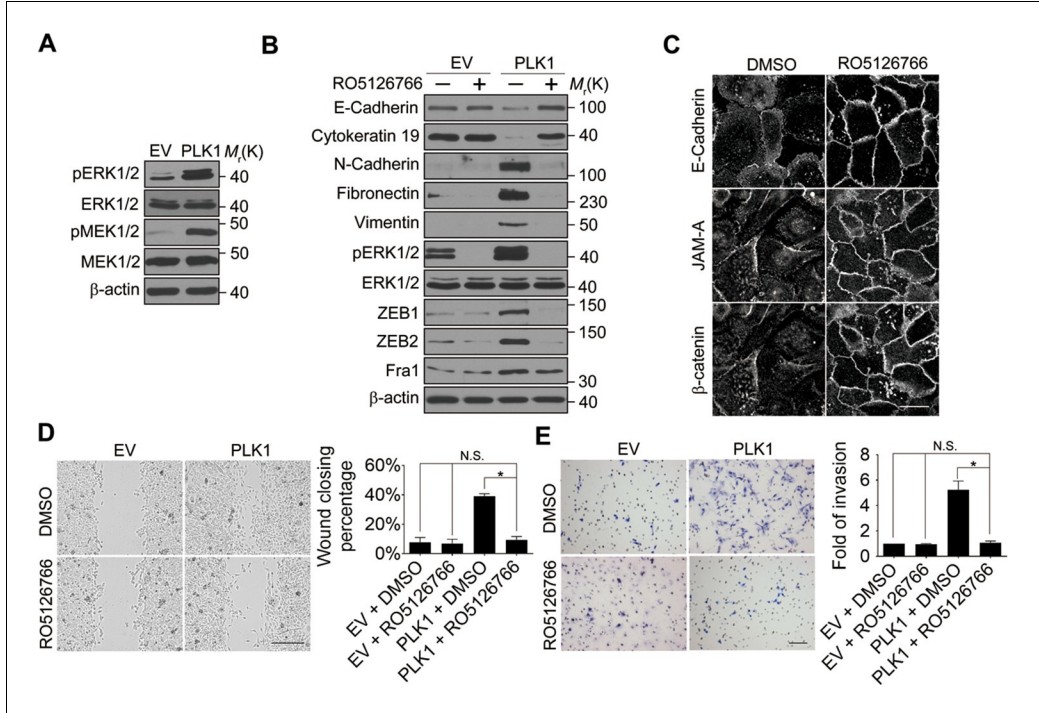

**Figure 5.** ERK1/2 activation is essential for PLK1-induced EMT and increased motility of prostate epithelial cells. (A) The effect of *PLK1* overexpression on activation (phosphorylation) of ERK1/2 and MEK1/2 was determined by immunoblotting. (B−E) Control (EV) and PLK1-overexpressing (PLK1) RWPE-1 cells were treated for 24 hr with either vehicle (DMSO) or RO5126766 (10 mM). The effects of MEK inhibition on PLK1-dependent induction of EMT (B), junctional disassembly (C), accelerated planar cell migration (D), and Matrigel invasion (E) were determined as described in *Figures 1* and *2*. The data are presented as the mean ± s.e.m. *p<0.05, two-tailed Student's *t*-test. NS: no significant difference. Scale bar, 20 μm (C), 500 μm (D), and 100 μm (E).

cells (*Figure 5D,E*). These results strongly support the role of the MEK/ERK pathway in PLK1-induced EMT and cell motility.

What is the molecular link between ERK activation and upregulation of EMT inducers such as ZEB1 and ZEB2? Recent studies have uncovered Fra-1 as an essential player in the ERK/ZEB signaling pathway, which induces EMT and cell migration (*Shin et al., 2010*). Fra1 belongs to the *Fos* gene family, whose protein products can dimerize with protein of the JUN family, thereby forming the transcription factor complex AP-1. Ectopic expression of Fra1 in epithelioid cells resulted in morphologic changes that resemble fibroblastoid conversion, and increased motility and invasiveness (*Kustikova et al., 1998*). Fra1 has been implicated as a potent regulator of anti-apoptosis, cell motility, and invasion in a variety of tumor cell types (*Milde-Langosch, 2005*; *Young and Colburn, 2006*). Based on these data, we sought to elucidate if Fra1 is essential for PLK1-induced EMT and accelerated cell motility in PCa. Overexpression of PLK1 in RWPE-1 cells (*Figure 6A,B*) and PrEC cells (*Figure 4—figure supplement 1*) resulted in increased Fra1 expression at both the mRNA and protein levels compared to control cells. Moreover, knocking down Fra1 in RPWE-1–PLK1 cells significantly decreased ZEB1/2 expression and reversed the biochemical signature of the EMT (*Figure 6C*). Importantly, Fra1 depletion reversed the increased wound healing and Matrigel invasiveness and restored assembly of intercellular junctions in PLK1-overexpressing RWPE-1 cells (*Figure 6D–F*). These data highlight Fra1 as a crucial molecular effector connecting ERK activation and induction of ZEB1/2 expression in the PLK1-induced signaling pathway leading to EMT and increased motility of PCa cells.

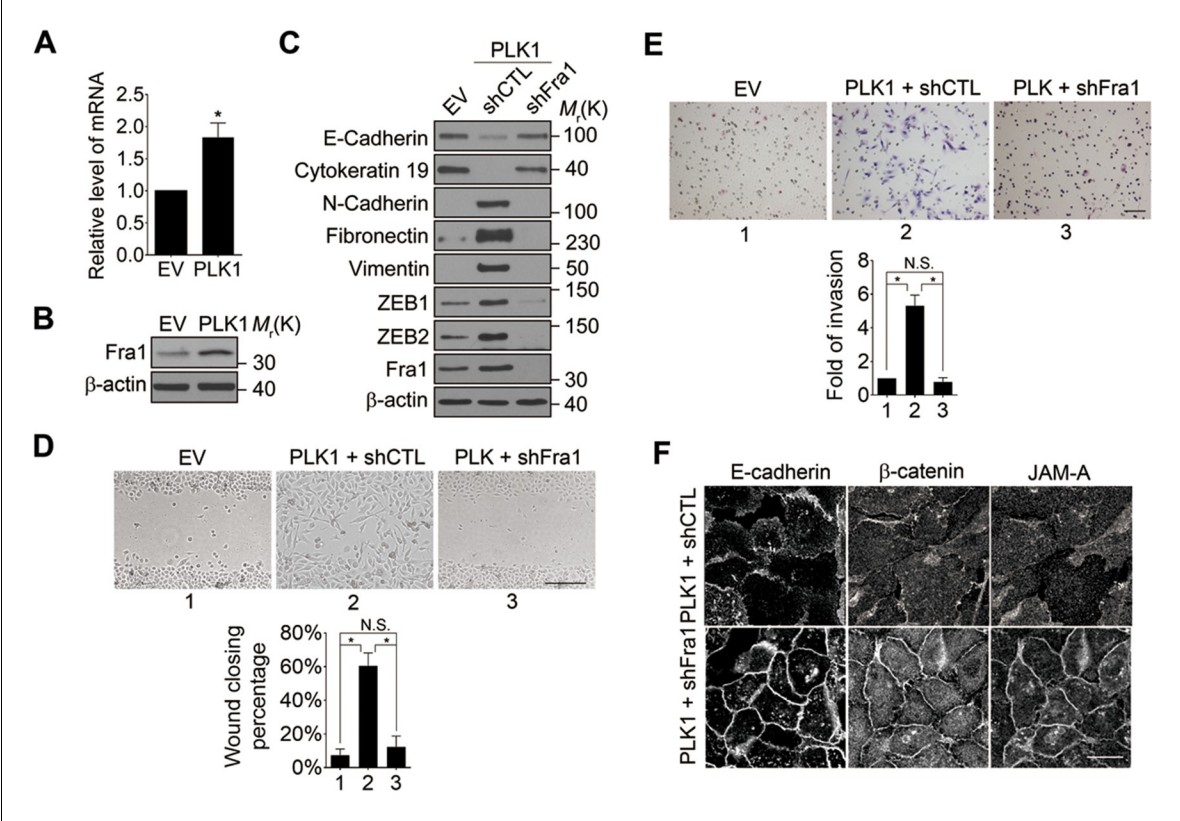

**Figure 6.** Fra1 is a critical mediator of PLK1-induced EMT and accelerated motility of prostate epithelial cells. (**A**, **B**) mRNA and protein expression of Fra1 were examined in control (EV) and PLK1-overexpressing (PLK1) RWPE-1 cells by real-time RT-PCR (**A**) and immunoblotting (**B**), respectively. The data are presented as the mean ± s.e.m. *p<0.05, two-tailed Student's *t*-test. (**C–F**) The effects of Fra1 depletion on PLK1-induced EMT (**C**), planar cell migration (**D**), Matrigel invasion (**E**), and epithelial junctional disassembly (**F**) were determined as described in *Figures 1* and *2*. NS: no significant difference. Scale bar, 500 µm (**D**), 100 µm (**E**), and 20 µm (**F**).

## PLK1-mediated phosphorylation activates CRAF to trigger downstream ERK1/2-Fra1-ZEB1/2 signaling

We next investigated the upstream events that could potentially mediate PLK1-dependent activation of MEK/ERK signaling. The RAF family of serine/threonine protein kinases consists of 3 isoforms, ARAF, BRAF, and CRAF, all of which serve as upstream kinases that evoke a serine/threonine phosphorylation cascade through sequential phosphorylation of MEK1/2, ERK1/2, and further downstream effectors to elicit a variety of cellular responses (*Kolch, 2000*). CRAF is ubiquitously expressed in mammalian cells, whereas ARAF and BRAF display more tissue-specific expression (*Hagemann and Rapp, 1999*). Activation of CRAF has served as a framework for the other 2 isoforms.

CRAF is the cellular proto-oncogene homologue of v-RAF, a retroviral oncogene, and is a central component of the MAP kinase cascade (*Heidecker et al., 1990*; *Morrison and Cutler, 1997*). Oncogenic CRAF causes EMT and invasion (*Hou et al., 2014*; *Lan et al., 2004*). To determine the involvement of CRAF in PLK1-induced EMT and accelerated motility of prostate epithelial cells, we determined if CRAF is activated in RWPE-1–PLK1 cells. It has been reported that phosphorylation of CRAF at S338 and S339 is essential for its activation and that the first essential role of CRAF kinase activity is to autophosphorylate S621 (*Diaz et al., 1997*; *Noble et al., 2008*). S621 phosphorylation plays a critical role in preventing CRAF proteasome-mediated degradation (*Noble et al., 2008*). Immunoblotting analysis with antibodies specifically recognizing those 2 different phosphorylated sites (S338 and S339) demonstrated a significant increase in CRAF phosphorylation following *PLK1* overexpression (*Figure 7A*). Furthermore, CRAF autophosphorylation of S621 was also significantly elevated upon *PLK1* overexpression (*Figure 7A*). Downregulation of CRAF expression by a specific

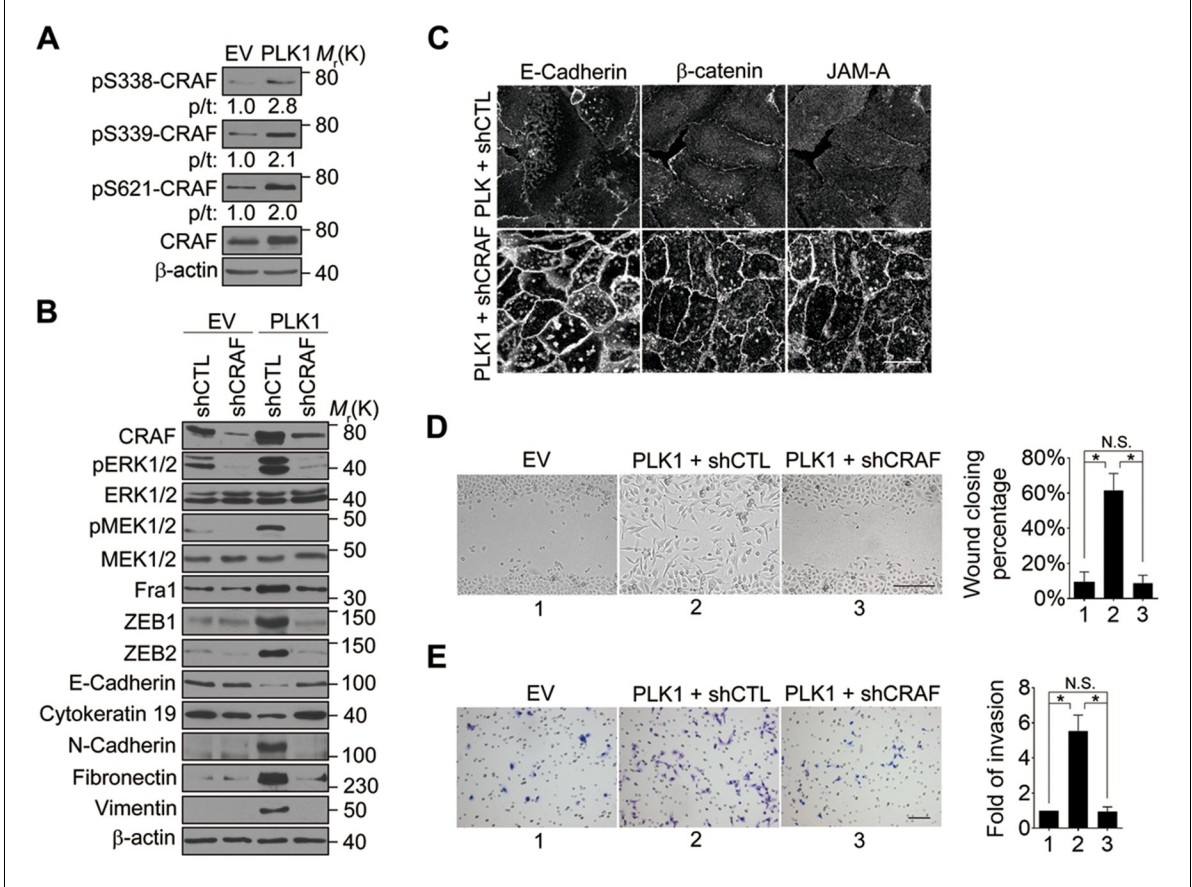

**Figure 7.** CRAF plays an essential role in PLK1-induced EMT and accelerated motility of prostate epithelial cells. (A) The effects of *PLK1* overexpression on the levels of total and phosphorylated CRAF were determined by immunoblotting. p/t: indicates densitometric intensity ratio of phosphorylated to total CRAF. (B–E) The effect of shRNA-mediated CRAF knockdown on PLK1-induced EMT (B), epithelial junctional disassembly (C), planar cell migration (D), and Matrigel invasion (E) were determined as described in *Figures 1–2*. N.S.: no significant difference. Scale bar, 20 µm (C), 500 µm (D), and 100 µm (E).

shRNA caused several prominent biochemical alterations in PLK1-overexpressing RWPE-1 cells. These alterations include inhibition of ERK signaling manifested by decreased MEK1/2 and ERK1/2 phosphorylation; decreased expression of Fra1, ZEB1, and ZEB2; and complete reversal of EMT-like expressional reprogramming of prostate epithelial cells (*Figure 7B*). Moreover, CRAF depletion restored assembly of intercellular junctions and suppressed wound healing and Matrigel invasion of PLK1-overexpressing RWPE-1 cells (*Figure 7C–E*). These results demonstrate that CRAF plays an essential role in PLK1-induced EMT and cell motility.

We next investigated the molecular mechanism by which PLK1 activates CRAF. The sequence surrounding S339 (residue 337–340: DS**S**Y) on CRAF resembles the PLK1 consensus phosphorylation sequence D/E-X-S/T-ψ (X denotes any amino acid and ψ denotes a hydrophobic amino acid) (*Nakajima et al., 2003*). We hypothesized that CRAF is a substrate of PLK1, and that PLK1-mediated phosphorylation of CRAF leads to its activation. To test these hypotheses, we first examined whether PLK1 interacts with CRAF. Endogenous PLK1-CRAF complex was detected in HeLa cells by immuno-precipitation, and PLK1-CRAF interaction appeared to be enhanced during the M phase of the cell cycle when PLK1 activity is maximal (*Figure 8A*). A GST pull-down assay was performed and showed CRAF association with GST-PBD, but not with GST or the GST-PBD mutant (*Figure 8B*), thereby suggesting that CRAF is a potential substrate for PLK1.

An *in vitro* kinase assay demonstrated that PLK1 directly phosphorylated CRAF at S338 and S339, but not at S621 (*Figure 8C*). Phosphorylation of CRAF at S338, S339, and S621 was significantly elevated in cells expressing WT or TD PLK1, but not KM PLK1 (*Figure 8D*). Cells expressing TD PLK1

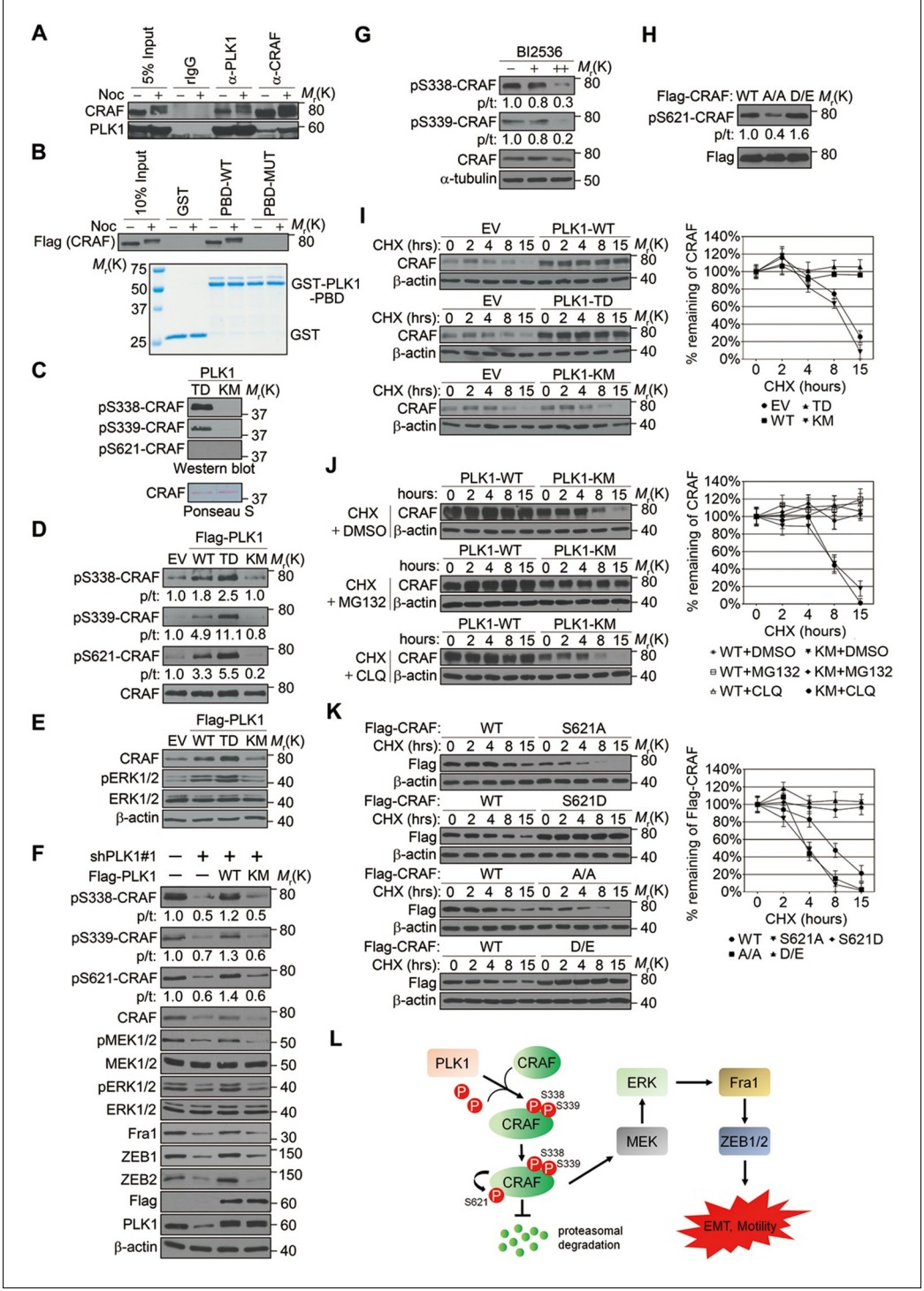

**Figure 8.** PLK1-mediated phosphorylation of CRAF leads to CRAF activation and stabilization. (**A**) Co-immunoprecipitation of endogenous PLK1 and CRAF from HeLa cell lysates obtained under control conditions or following nocodazole treatment to arrest cells at M phase. (**B**) GST pull-down assay was performed using GST-tagged wild-type (WT) or mutant (MUT) PLK1 PBD (upper panel). Equal loading of kinase substrates is indicated by Coomassie Blue staining (lower panel). (**C**) Bacterially expressed CRAF was subjected to *in vitro* kinase assays with constitutively active (TD) or kinase-defective (KM) PLK1 mutants purified from insect cells. The phosphorylation of CRAF was observed by immunoblotting with the indicated antibodies. Ponseau S staining of the blot was used to indicate equal loading of the assays. (**D**) RWPE-1 cells were infected with lentivirus expressing empty vector (EV), wild-type PLK1 (WT), constitutively active T210D (TD), or kinase-dead K82M (KM) mutants. Total cell lysates were

*Figure 8 continued on next page*

*Figure 8 continued*

subject to immunoprecipitation with CRAF antibody and then analyzed by immunoblotting. p/t: indicates densitometric intensity ratio of phosphorylated to total CRAF. (**E**) Immunoblotting analysis of ERK1/2 activation (phosphorylation) in RWPE-1 cells expressing EV or PLK1 WT, TD, or KM mutants. (**F**) C4-2B cells were infected with lentiviral shRNA constructs that target either the 3'-UTR of endogenous PLK1 (shPLK1#1) or serve as a control. Wild-type (WT) or kinase-defective (KM) PLK1 were then re-expressed in PLK1 knockdown cells. The levels of phosphorylated and total CRAF, MEK and ERK, Fra1, and ZEB1/2 were determined by Western blotting analysis. (**G**) C4-2B cells were arrested in M phase by nocodazole (0.1 mg/mL) treatment for 16 hr, and then subject to BI 2536 treatment for 30 min. The levels of phosphorylated and total CRAF were determined by Western blotting analysis. (**H**) Flag-tagged CRAF WT-, CRAF-A/A-, and CRAF-D/E-overexpressing RWPE-1 cells were subjected to immunoprecipitation with anti-Flag antibody followed by immunoblotting with the indicated antibodies. (**I**) Control- (EV), PLK1 WT-, PLK1 TD- and PLK1 KM-overexpressing RWPE-1 cells were treated with cycloheximide (CHX: 20 µg/ml) for the indicated times, and the CRAF protein levels were monitored by immunoblotting (left panel). Quantification of the endogenous CRAF protein levels relative to β-actin expression is shown in the right panel. The data are presented as the mean ± s.e.m. (**J**) PLK1 WT-, and PLK1 KM-overexpressing RWPE-1 cells were treated with cycloheximide (CHX: 20 µg/ml) along with vehicle (DMSO), MG132 (2.5µM), or chloroquine (CLQ, 50 mM) for the indicated times, and level of CRAF protein was examined by immunoblotting. (**K**) Flag-tagged CRAF WT-, CRAF-S621A-, CRAF-S621D-, CRAF-A/A-, and CRAF-D/E-overexpressing RWPE-1 cells were treated with cycloheximide (CHX: 20 µg/ml) for the indicated time and the level of CRAF protein was determined by immunoblotting. (**L**) A proposed diagram of a novel signaling cascade that mediates PLK1-dependent induction of EMT and cell motility.

The following figure supplement is available for figure 8:

**Figure supplement 1.** Downregulation of PLK1 led to dramatic reduction of CRAF phosphorylation at S338, S339, and S621 that was rescued by WT PLK1, but not KM PLK1.

displayed higher levels of phosphorylated CRAF compared to those with WT PLK1 (*Figure 8D*). These data suggest that the enzymatic activity of PLK1 is both directly and indirectly responsible for those phosphorylation events. Consistently, phosphorylated ERK was significantly increased in cells expressing WT or TD PLK1, but not KM PLK1 (*Figure 8E*), suggesting that a PLK1-induced phosphorylation event triggers the activation of ERK signaling. On the other hand, shRNA-mediated downregulation of PLK1 led to dramatic reduction of CRAF phosphorylation at S338, S339, and S621, which was rescued by WT PLK1, but not KM PLK1 (*Figure 8F* and *Figure 8—figure supplement 1*). Furthermore, inhibition of ERK signaling, manifested by decreased MEK1/2 and ERK1/2 phosphorylation along with reduced expression of Fra1, ZEB1, and ZEB2, was observed upon PLK1 knock down, and these effects were reversed by re-expressing WT, but not KM PLK1 (*Figure 8F* and *Figure 8—figure supplement 1*). To further confirm that PLK1 directly phosphorylates CRAF at S338 and S339, cells were synchronized in M phase by nocodazole treatment followed by a transient inhibition of PLK1 activity with 2 different concentrations of BI 2536 for 30 min. Significant reduction in phosphorylation of CRAF at S338 and S339 was immediately observed upon treatment with the higher dose of the PLK1 inhibitor (*Figure 8G*), thereby suggesting that PLK1 indeed directly phosphorylates CRAF at S338 and S339. Furthermore, when we blocked the PLK1 phosphorylation sites on CRAF by mutating S338 and S339 to alanines (CRAF-A/A), S621 phosphorylation was significantly diminished as compared to WT CRAF or the phospho-mimetic mutant (CRAF-D/E) (*Figure 8H*), suggesting that PLK1-induced phosphorylation of CRAF at S338 and S339 contributes to the CRAF autophosphorylation of S621. Taken together, these findings demonstrate that PLK1 is responsible for phosphorylation-mediated activation of CRAF, which leads to CRAF autophosphorylation.

Elevated CRAF protein levels, but unchanged CRAF mRNA expression, were observed in RWPE-1–PLK1 cells (*Figure 7A*, data not shown). In order to elucidate the mechanisms of this selective upregulation of CRAF protein, we asked whether PLK1 can stabilize CRAF. Cells were treated with cycloheximide to inhibit *de novo* protein synthesis and the expression level of CRAF was compared at different time points (0, 2, 4, 6, 8, and 15 hr) in control cells and cells overexpressing different forms of PLK1 (WT, TD, and KM). Immunoblotting analysis demonstrated significant degradation of CRAF in control cells and cells expressing an inactive PLK1-KM mutation (*Figure 8I*). CRAF

degradation was inhibited by MG132 (a proteasome inhibitor), but not by chloroquine (a lysosomal inhibitor) (*Figure 8J*). In contrast, no significant CRAF degradation was observed in cells overexpressing either WT PLK1 or its constitutively active TD mutant (*Figure 8I*). Given the previous finding that CRAF autophosphorylation of S621 is critical for preventing its degradation (*Noble et al., 2008*) and our observations that PLK1 phosphorylation of CRAF leads to its activation and subsequent autophosphorylation at S621 (*Figure 8A–H*), we hypothesized that PLK1-induced phosphorylation of CRAF contributes to CRAF stabilization by promoting its autophosphorylation. To corroborate the previous findings from another group (*Noble et al., 2008*), we mutated S621 of CRAF to alanine (CRAF-S621A) or aspartate (CRAF-S621D) and examined their stability. Consistently, CRAF-S621A displayed a much shorter half-life when compared to WT CRAF or CRAF-S621D (*Figure 8K*). Importantly, the CRAF-A/A mutant (lacking PLK1-mediated phosphorylation) was quickly degraded over a 15-h time course (*Figure 8K*), which is in agreement with our observation that the CRAF-A/A mutant harbors a much lower level of S621 phosphorylation (*Figure 8H*). In sharp contrast, the CRAF WT and the CRAF-D/E mutant showed minimal degradation during that period of time (*Figure 8K*). These results demonstrate that PLK1 increases the stability of CRAF protein by preventing proteasome degradation.

## Discussion

PLK1 is an established mitotic kinase that is involved in multiple steps of mitotic progression. The present study revealed unanticipated non-canonical functions of PLK1 as a potent inducer of EMT and a stimulator of the motile phenotype of prostate epithelial cells. These functions of PLK1 are mediated by the CRAF-MEK1/2-ERK1/2-Fra1-ZEB1/2 signaling pathway and are independent of effects on cell cycle regulation.

Our results are consistent with previous studies that suggested pro-migratory activity of PLK1 in colorectal, breast, thyroid cancer, and melanoma (*Han et al., 2012*; *Kneisel et al., 2002*; *Rizki et al., 2007*; *Zhang et al., 2012*). However, our study is the first to demonstrate the causal role of PLK1 in regulating cancer cell motility. Indeed, by using a combination of gain- and loss-of-function approaches, we established that PLK1 is necessary and sufficient to promote both planar migration and invasion of prostate epithelial cells (*Figures 1* and *2*). Furthermore, our time-lapse microscopy experiments clearly demonstrated that PLK1 directly regulates the velocity of epithelial cell migration, independently of its effects on other cellular processes (*Figures 1D–F*).

PLK1 controls prostate epithelial cell motility by mechanisms requiring transcriptional reprogramming and dramatic alterations of cell phenotype. Indeed, overexpression of PLK1 induced EMT-like alterations in prostate epithelial cells (*Figure 2*), whereas PLK1 knockdown restored epithelial features of invasive prostate cancer cell lines (*Figure 3*). This PLK1-dependent regulation of cell phenotype has not been previously reported in any cell type, and it is likely to underlie the observed effects of PLK1 on epithelial cell motility. There are several potential mechanisms by which the induction of EMT can promote cell motility. One mechanism involves disassembly of epithelial junctions that weaken intercellular adhesions, thereby allowing cell dissemination (*Godde et al., 2010*; *Le Bras et al., 2012*). Another mechanism involves rearrangement of the actomyosin cytoskeleton from epithelia-specific perijunctional bundles to basal stress fibers that are characteristic of mesenchymal cells. This rearrangement enhances cell-matrix adhesion and enables more efficient cell migration (*Martin et al., 2014*; *Yilmaz and Christofori, 2009*). We observed both junctional disassembly and cytoskeletal rearrangements in prostate epithelial cells undergoing PLK1-mediated EMT (*Figure 2*). Our finding that *PLK1* overexpression potently induced EMT in prostate epithelium is important given recent evidence highlighting EMT as an emerging mechanism of PCa progression, metastasis, and therapeutic resistance. For instance, several EMT markers, such as vimentin and N-cadherin, are commonly expressed in circulating tumor cells from patients with relapsed metastatic PCa (*Armstrong et al., 2011*). A 'cadherin switch' with increased N-cadherin and reduced E-cadherin expression has been associated with relapse after PCa surgery and development of metastatic disease (*Gravdal et al., 2007*; *Umbas et al., 1992*). Work from Reiter's group demonstrates a central role for N-cadherin in PCa metastasis (*Tanaka et al., 2010*). Finally, the levels of ZEB proteins positively correlate with Gleason grading and PCa metastasis (*Graham et al., 2008*).

Through a series of biochemical analyses, we delineated the molecular mechanism underlying PLK1-mediated cell motility and EMT. Previous studies demonstrated that CRAF-MEK1/2-ERK1/2

signaling plays a crucial role in the regulation of EMT as well as cell migration and invasion in several cancers (*Birchmeier et al., 1993*; *Hay and Zuk, 1995*; *Schoenenberger et al., 1991*). The CRAF-MEK1/2-ERK1/2 pathway is regulated by Ras as well as various kinases including Src, PKC, and PAK (*King et al., 1998*). In this study, we discovered a novel regulatory pathway for MAPK signaling. We demonstrated that CRAF is a physiological substrate of PLK1 (*Figure 8*). CRAF consists of a N-terminal regulatory domain and a C-terminal catalytic domain. The 'N-region', located at the N-terminal of the kinase domain, contains the critical activating phosphorylation sites, S338 and S339 (*Diaz et al., 1997*; *Edin and Juliano, 2005*). S338 can be phosphorylated by the p21 activated kinase (PAKs) (*King et al., 1998*; *Chaudhary et al., 2000*; *Sun et al., 2000*) and other as yet unidentified kinases (*Chiloeches et al., 2001*). Our data support a dual-mechanism model of PLK1-mediated regulation of CRAF signaling (*Figure 8L*): PLK1 directly interacts with and phosphorylates CRAF at S338 and S339, which leads to CRAF activation. The activated CRAF undergoes autophosphorylation of S621, thereby preventing proteasome-mediated degradation of CRAF, which generates a positive-feedback loop, leading to a further increase in CRAF level and activity. This activation event triggers the activation of downstream MEK1/2-ERK1/2 signaling in prostate epithelial cells overexpressing PLK1 (*Figure 8L*). Consistently, Mielgo et al. reported that PLK1 associates with CRAF and that this interaction does not require CRAF kinase activity (*Mielgo et al., 2011*). Interestingly, they also found that CRAF indirectly promotes PLK1 activation (*Mielgo et al., 2011*). This adds an additional layer of complexity to the PLK1-CRAF interplay to further activate CRAF. In addition, PLK1 may differentially trigger distinct signaling pathways under different physiological conditions. For instance, PLK1 can directly activate MEK/ERK signaling through phosphorylation of the MEK activating site in airway smooth muscle cells (*Jiang and Tang, 2015*), although that is not the case for PCa (*Figure 8*).

We showed that the ERK1/2-Fra1-ZEB1/2 pathway is aberrantly activated in PLK1-overexpressing prostate epithelial cells and that blocking the ERK/Fra1/ZEB1/2 pathway by shRNA or pharmacological inhibition reverses EMT and decreases cell motility in these cells (*Figures 4–6*). These results indicate that PLK1 promotes EMT and motility through the CRAF-MEK1/2-ERK1/2-Fra1-ZEB1/2 pathway in prostate epithelial cells. These findings highlight PLK1 as a critical molecular rheostat that controls phenotypic plasticity of normal prostate epithelial cells as well as PCa progression and dissemination. Identification of the CRAF-MEK1/2-ERK1/2 signaling cascade as an essential downstream effector of PLK1 in prostate epithelial cells corresponds with the known roles of this cascade in the pathogenesis of various types of cancer, including PCa (*McCubrey et al., 2007*; *Roberts and Der, 2007*). Given the fact that the frequency of *Ras* gene mutations in prostate tumors is low (*Carter et al., 1990*), PLK1-induced activation of CRAF provides mechanistic insights into the aberrant activation of CRAF-MEK1/2-ERK1/2 signaling frequently detected in PCa. Future experiments will further establish PLK1-mediated regulation of CRAF-MEK1/2-ERK1/2 signaling in patients with PCa. Interestingly, it was reported that ERK2/Fra1/ZEB1/2 signaling is responsible for induction of EMT in RasG12V-transformed MCF10A cells (*Shin et al., 2010*), suggesting that ERK/Fra1/ZEB signaling could be a common pathway to induce EMT in mammalian epithelial cells.

PLK1 deregulation has been linked with the initiation and progression of many human cancers, including PCa (*Cholewa et al., 2013*; *Takai et al., 2005*; *Weichert et akl., 2004*). The current dogma in the field implies that PLK1 controls cancer development through multiple mechanisms, including canonical regulation of mitosis and cytokinesis as well as modulation of DNA replication and cell survival (*Deeraksa et al., 2013*; *Luo and Liu, 2012*). Whether and how PLK1 drives PCa metastasis *in vivo* remain to be elucidated. Our xenograft studies showed that *PLK1* overexpression in human prostate epithelial cells leads to cellular transformation *in vitro* and promotes tumor formation in NSG mice, which suggests that PLK1 has a tumor-promoting role in the prostate. Strikingly, NSG mice engrafted with RWPE-1–PLK1 cells developed lung micrometastases at a high frequency (*Figure 1—figure supplement 2*), which provides *in vivo* evidence that PLK1 plays a role in PCa invasion and metastasis. Our study establishes the role of PLK1 in the induction of EMT and stimulation of cell motility, which adds a novel and previously unanticipated role for PLK1 during PCa development. Furthermore, our observation that partial downregulation of PLK1 in metastatic PCa cells has no effect on cell cycle progression and cell viability suggests that a low level of PLK1 is sufficient to maintain cell viability and regulate the cell cycle. In contrast, upregulation of PLK1 promotes its non-canonical functions, including stimulation of EMT and acceleration of cell motility. This could explain

why normal cells develop a complex system to tightly control the level and activity of PLK1 throughout the cell cycle, and how increased PLK1 promotes tumorigenesis.

EMT is an important mechanism of tumor progression and metastasis (*Kalluri and Weinberg, 2009*; *Yang and Weinberg, 2008*). Loss of cell-cell contacts and reorganization of the intracellular cytoskeleton during EMT results in increased cell migration and invasion (*Moreno-Bueno et al., 2008*). This allows cells to invade the surrounding stroma and vasculature, which leads to tumor dissemination and metastases (*Hugo et al., 2007*). However, the mechanisms that drive EMT in PCa remain elusive (*Gravdal et al., 2007*; *Howard et al., 2008*). This study provides strong evidence that PLK1 is a key regulator of EMT in prostate epithelial cells and in PCa, which represents an underlying mechanism for PLK1-driven PCa progression and spreading. EMT also enables cancer cells to avoid apoptosis, anoikis, and oncogene addiction (*Jordan et al., 2011*; *Tiwari et al., 2012*). An emerging mechanism of EMT is reprogramming epithelial cells into cancer stem cells that have been strongly associated with resistance to chemotherapy and disease recurrence (*Chang et al., 2013*). Whether PLK1-induced EMT contributes to these oncogenic processes will need further investigation. Interestingly, the low level of endogenous PLK1 observed in RWPE-1 cells appears to be sufficient to mediate growth-factor–induced EMT (*Figure 3—figure supplement 4*). This suggests that in tumor cells that do not overexpress PLK1, the low level of PLK1 that is required for cell division could also mediate induction of EMT under certain circumstances, such as in the tumor microenvironment. This finding would have important implications in targeting PLK1 for anticancer treatments. For instance, targeting PLK1 might be applied to a wide variety of cancers, regardless of their PLK1 expression levels, to prevent induction of EMT and tumor metastasis. Including PLK1 inhibitors in current standard therapeutic regimens could expand the scope of clinical efficacy that currently available drugs have established.

In conclusion, we demonstrate for the first time that PLK1 drives planar cell migration and matrix invasion of prostate epithelial cells and PCa by mechanisms involving induction of EMT. This previously unanticipated pro-migratory activity of PLK1 is driven by direct phosphorylation and activation of CRAF, which results in enhanced signaling through the MEK1/2-ERK1/2-Fra1-ZEB1/2 pathway. The insights gained from this study will fundamentally advance our understanding of the oncogenic functions of PLK1 and the molecular basis of PCa development and metastatic dissemination, which have the potential to facilitate optimization of treatment regimens targeting PLK1 signaling to significantly enhance anticancer efficacy. Furthermore, our novel findings may be generalizable to many other cancers since preliminary clinical evidence suggests that PLK1 is involved in the development of several other human cancers.

## Materials and methods

### Cell lines, synchronization, and reagents

The cell lines used were as follows: RWPE-1 (prostate epithelial cells); LNCaP, LAPC4, PC3, C4-2B, and DU145 (prostate cancer cells); 293T (human embryonic kidney cells); HeLa (human cervical adenocarcinoma cells, obtained from ATCC, Manassas, VA); PrEC (human primary prostate epithelial cells, obtained from Lonza, Basel, Switzerland); and ARCaP$_E$ and ARCaP$_M$ (Novicure Biotechnology, Birmingham, AL). All cell lines were tested and found to be free of mycoplasma and were cultured for no more than 10 passages according to the manufacturer's recommendations. All cell types were checked for proper morphology prior to every experiment and consistently monitored for changes in cell replication that might suggest Mycoplasma contamination. Cell synchronization was performed as described previously (*Yuan et al., 2014*). The inhibitors used were as follows: RO5126766 (Active Biochem, Maplewood, NJ), nocodazole (Sigma-Aldrich, St. Louis, MO), BI2536 (Selleck Chemicals, Houston, TX), cycloheximide (Sigma-Aldrich), MG132 (Sigma-Aldrich), chloroquine (Sigma-Aldrich), recombinant human TGF-β1 (Thermo Fisher Scientific, Grand Island, NY), and recombinant human EGF (BD Biosciences).

### Plasmids, small-interfering RNAs (siRNAs), and small hairpin RNAs (shRNAs)

Baculovirus encoding human WT *PLK1* and kinase inactive (K82M, KM) mutants were generous gifts from Dr. R.L. Erikson (Harvard University, Boston, MA). Flag-PLK1 WT, the constitutively active form

(TD) and KM were subcloned into the pLVX-AcGFP-N1 lentiviral vector (Clontech, Mountain View, CA). Human full-length *PLK1* WT and KM mutant in the pRc/CMV vector were a gift from Dr. E.A. Nigg (Max Planck Institute of Biochemistry). pGEX-4T-1-PLK1-PBD WT and MUT (mutant, H538A/K540M) were kindly provided by Dr. MB Yaffe (Massachusetts Institute of Technology, Cambridge, MA). pcDNA 3.0-Flag-CRAF was a kind gift from Dr. KC Yeung (University of Toledo, Toledo, OH). The fragment of CRAF spanning 275-648aa was cloned into pET-28a (+). MISSION lentiviral shRNAs for PLK1 (#1: TRCN0000121072; Clone ID: NM_005030.3-1893s1c1 and #2: TRCN0000121323; Clone ID: NM_005030.3-1073s1c1), ZEB1 (TRCN0000017565; Clone ID: NM_030751.2-572s1c1), ZEB2 (TRCN0000013530; Clone ID: NM_014795.2-2202s1c1), Fra1 (TRCN0000019539; Clone ID: NM_005438.2-780s1c1), and CRAF (TRCN0000001068, Clone ID: NM_002880.x-1529s1c1) were from Sigma-Aldrich. siRNA duplexes against human PLK1 (access no. J-003290-09-0005 and J-003290-10-0005), and negative control siRNA (accession no. D-001810-10-05) were purchased from Dharmacon Research.

## Lentivirus preparation and infection

The cell line overexpressing PLK1 was produced by a third-generation lentiviral system (pLVX-AcGFP-N1, pMDLg/pRRE, pRSV-Rev, and pMD2.G). The 2 shRNAs were produced by a second-generation lentiviral system (pLKO.1, psPAX2, and pMD2.G). Briefly, a lentiviral vector containing the shRNA or a transgene along with packing vector and envelope vector were cotransfected into 293T cells using Lipofectamine 2000 (Thermo Fisher Scientific) according to the manufacturer's instructions. The supernatants containing virus particles were collected, filtered, and concentrated. Suspended virus was applied to target cells with 8 µg/mL polybrene. The cells were selected in the presence of puromycin (2 µg/mL for prostate cancer cell lines; 0.5 µg/mL for RWPE-1 and PrEC cells) 72 hr after infection.

## Transfection

Transient transfections of ARCaP$_E$ cells with a control or a PLK1 expression vector were performed using Lipofectamine 2000 per the manufacturer's protocol. For siRNA depletion experiments, ARCaP$_M$ cells were transfected with these siRNA duplexes using DharmaFECT1 siRNA transfection reagent following the manufacturer's instructions.

## Antibodies, immunoblotting, and immunoprecipitation

The antibodies used in this study were: anti-PLK1 (clone 36–298, Thermo Fisher Scientific, 1:1000), anti–phospho-PLK1 (T210) (no. 5472, Cell Signaling Technology, Danvers, MA, 1:500), anti–Flag (F3165, M2, Sigma-Aldrich, 1:5000), anti–β-actin (SAB4200248, Sigma-Aldrich, 1:5000), anti-E-cadherin (no. 610181, BD Biosciences, Franklin Lakes, NJ, 1:2000), anti-N-cadherin (ab12221, Abcam, Cambridge, MA, 1:1000), anti–Vimentin (V6389, Sigma-Aldrich, 1:1000), anti–Cytokeratin 19 (GTX27754, GeneTex, Irvine, CA, 1:1000), anti–SM22α (ab10135, Abcam, 1:1000), anti–Fibronectin (F3648, Sigma-Aldrich, 1:2000), anti–ERK (no. 4695, Cell Signaling Technology, 1:500), anti–phospho-ERK (Thr202/Tyr204) (no. 4370, Cell Signaling Technology, 1:500), anti–MEK1/2 (no. 9122S, Cell Signaling Technology, 1:500), anti–phospho-MEK1/2 (Ser217/221) (9121S, Cell Signaling Technology, 1:500), anti–CRAF (sc-133, Santa Cruz Biotechnology, Dallas, TX, 1:500), anti–phospho-CRAF (S338) (9427, Cell Signaling Technology, 1:500), anti–phospho-CRAF (S339) (Bs-5652R, Bioss, Wobum, MA, 1:250), anti–ZEB1 (no. 5825, ProSci, 1:250), anti–ZEB2 (AP12086b, ABGENT, San Diego, CA, 1:250), anti–Fra1 (sc-28310, Santa Cruz Biotechnology, 1:500), anti–Snail (no. 4719, Cell Signaling Technology, 1:500), and anti–PSA (sc-7316, Santa Cruz Biotechnology, 1:1000). Cells were lysed in RIPA buffer (50 mM Tris-HCl [pH 7.5], 150 mM NaCl, 0.1% SDS, 1% Nonidet P-40, 1% Na-deoxycholate, and 1 mM EDTA) containing protease and phosphatase inhibitors for 20 min at 4°C. Cell lysates were centrifuged for 10 min at 4°C. For immunoprecipitation or co-immunoprecipitation experiments, cleared lysates (1–2 mg protein) were immunoprecipitated with the indicated polyclonal antibodies and protein A-Sepharose for 1 hr. Proteins were separated on a 7.5% SDS-PAGE gel, transferred to Immobilon P membranes, and immunoblotted with the indicated antibodies.

## Immunofluorescence labeling and confocal microscopy

In order to label AJ/TJ and cytoskeletal proteins, cells were fixed with 100% methanol for 20 min at -20°C. Fixed cell monolayers were washed with HBSS, blocked with 1% bovine serum albumin in HBSS (blocking buffer) for 60 min at room temperature, and incubated for another 60 min with primary antibodies diluted in blocking buffer. Cells were then washed, incubated for 60 min with Alexa-Fluor−conjugated secondary antibodies, rinsed with blocking buffer, and mounted on slides with ProLong Gold Antifade Reagent (Thermo Fisher Scientific). Fluorescently-labelled cell monolayers were examined using a Zeiss LSM700 laser scanning confocal microscope (Zeiss Microimaging Inc., Thornwood, NY). The Alexa Fluor 488 and 555 signals were imaged sequentially in frame-interlaced mode to eliminate cross-talk between channels. The images were processed using Zen 2011 software and Adobe Photoshop. Images shown are representative of at least 3 experiments, with multiple images taken per slide. Antibodies used for immunofluorescence labeling: anti−E-cadherin (no 610181, BD Bioscience, 1:300), anti−ZO_1 (no 40–2200, Thermo Fisher Scientific, 1:300), anti−non-muscle myosin IIB (no. 3404, Cell Signaling Technology, 1:400), anti–β-catenin (C2206, Sigma-Aldrich, 1:600), and anti–JAM-A (clone J10.4, provided by Dr. Charles Parkos, University of Michigan, 1:200).

## Real-time qPCR analyses

Total RNA was isolated using TRIzol (Thermo Fisher Scientific) and reverse transcribed into cDNA using SuperScript III First-Strand Synthesis System (Life Technologies) according to the manufacturer's instructions. Real-time qPCR was performed using an ABI 7900HT thermal cycler and the FastStart Universal SYBR Green Master (Roche, Basel, Switzerland). The data were normalized to the amount of GAPDH transcript. The primer sequences were as follows: E-cadherin: 5'-ACAGCCCCGCCTTATGATT-3' (forward) and 5'-TCGGAACCGCTTCCTTCA-3' (reverse); Cytokeratin 19: 5'-GGTCATGGCCGAGCAGAA-3' (forward) and 5'-TTCAGTCCGGCTGGTGAAC-3' (reverse); N-cadherin: 5'-TGGGAATCCGACGAATGG-3' (forward) and 5'-GCAGATCGGACCGGATACTG-3' (reverse); Fibronectin: 5'-CATGAGACTGGTGGTTACATGTTAGA-3' (forward) and 5'-GCATGA TCAAAACACTTCTCAGCTA-3' (reverse); Vimentin: 5'-AATGACCGCTTCGCCAACT-3' (forward) and 5'- ATCTTATTCTGCTGCTCCAGGAA-3' (reverse); SM22α: 5'-GGCATGAGCCGCGAAGT-3' (forward) and 5'-TCCTCCAGCTCCTCGTCATACT-3' (reverse); SNAIL: 5'-ACCCCAATCGGAAGCC TAAC-3' (forward) and 5'-GCTGGAAGGTAAACTCTGGATTAGA-3' (reverse); SLUG: 5'- CAGC TACCCAATGGCCTCTCT-3' (forward) and 5'-GGACTCACTCGCCCCAAAG-3' (reverse); ZEB1: 5'-CAAATGTGGAAAGCGCTTCTC-3' (forward) and 5'-GTAGGAGTAGCGATGATTCATGTGTT-3' (reverse); ZEB2: 5'-CGCATTTCCCCCTGCTACT-3' (forward) and 5'-TGGTCGTAGCCCAGGAATAC TG-3' (reverse); E47 (TCF3): 5'-GCGGAACCTGAATCCCAAA-3' (forward) and 5'-CACACCTGA CACCTTTTCCTCTT-3' (reverse); Fra1: 5'-GCCGCCCTGTACCTTGTATC-3'; (forward) and 5'-CAG TGCCTCAGGTTCAAGCA-3' (reverse); TWIST: 5'-GGAGTCCGCAGTCTTACGAG-3' (forward) and 5'-TCTGGAGGACCTGGTAGAGG-3' (reverse); AR: 5'-TCACAGCCTGTTGAACTCTTC-3' (forward) and 5'-ACCTACTTCCCTTACCCCGCCT -3' (reverse); GAPDH: 5'-GAAATCCCATCACCATCTTCCA-3' (forward) and 5'-CCAGCATCGCCCCACTT-3' (reverse).

## Wound healing scratch assay

Cells were plated on collagen-I–coated slides. Once cells grew to confluence, a wound was introduced by scratching the confluent monolayer with a pipette tip. The images were acquired at 0, 24, and 48 hr post-wounding. The relative surface area travelled by the leading edge was calculated using TSscratch software (*Gebäck et al., 2009*). The figure shows representative images of 3 independent experiments performed in triplicate.

## Transwell invasion assays

The transwell invasion assay was performed using Matrigel Invasion Chambers (BD Bioscience) as we previously described (*Fu et al., 2003*). RWPE-1 cells ($2\times10^5$ cells/well), PrEC cells ($2\times10^5$ cells/well), or prostate cancer cell lines (PC3, DU125, and C4-2B; $5\times10^4$ cells/well) were plated into culture inserts coated with Matrigel.

## Time-lapse cell motility assay

Cells were sparsely plated on a collagen-coated, 35-mm culture dish and were imaged for 9 hr at 3-min time intervals. For all time-lapse recordings, the culture dish was placed in a microincubator to maintain proper environmental conditions (37°C, pH7.4). All images were acquired using a Zeiss Cell Observer Spinning Disc confocal microscope and analyzed using NIH ImageJ software. Velocity measurements and tracking diagrams were made using the manual tracking plugin for NIH ImageJ software and Adobe Photoshop. Thirty random cells per experiment were analyzed. The experiments were repeated 3 times.

## GST pull-down experiments

Glutathione-sepharose resin-coupled GST-PLK1 PBD (WT and MUT) or the GST control was incubated with cell lysates for 1 hr at 4°C. The resins were washed 4 times using NETN buffer (150 mM NaCl, 1 mM EDTA, 20 mM Tris pH 8, 0.5% NP-40). Resin-bound complexes were eluted by boiling, separated by SDS-PAGE, and analyzed by Western blotting.

## Cell cycle analysis by flow cytometry

Cells were collected, washed with PBS and fixed with ice cold 70% ethanol for at least 1 hr. Cells were then washed twice in PBS and treated for 30 min at 37°C with RNase A at 5 µg/mL and PI at 50 µg/mL, and analyzed on a FACScan flow cytometer (Becton Dickinson). The percentage of cells in different cell cycle phases was calculated using ModFit LT for Mac (BD Biosciences).

## Measure of apoptosis by Annexin-V and PI staining

Upon genetic modification of PLK1 expression (downregulation of endogenous PLK1 and re-expression of exogenous PLK1) in DU145 and C4-2B cells, the entire cell population was subjected to double staining for FITC-annexin V and PI using a FITC-annexin V apoptosis detection kit (BD Biosciences) and analyzed by flow cytometry for apoptotic events according to the manufacturer's instructions.

## MTS assay

Cells were plated in 96-well tissue culture plates at a density of $1 \times 10^3$ cells per well. The cell viabilities were assessed by means of a CellTiter 96 Aqueous One solution cell proliferationassay (Promega) accordingto the manufacturer's instructions. Experiments were performed in triplicate.

## Protein purification and *in vitro* kinase assay

Human PLK TD and KM were expressed in the baculovirus/insect cell system as we described previously (*Fu et al., 2008*). His-CRAF 275–648 aa fusion protein was expressed in the *Escherichia coli* BL21 strain. Kinase and substrates (CRAF proteins) were incubated in kinase buffer (20 mM Hepes, pH 7.4, 150 mM KCl, 10 nM MgCl$_2$, 1 mM EGTA, 0.5 mM dithiothreitol, 5 mM NaF, 100 µM ATP,) for 30 min at 30°C. Reactions were stopped by the addition of SDS sample buffer. Then samples were heated for 5 min to 95°C before analysis by SDS-PAGE and Western blot analysis with specific antibodies.

## Soft agar colony formation assay

Assays of colony formation in soft agar were performed using standard methods as we described previously (*Fu et al., 2003*). Cells were plated ($1 \times 10^4$ cells/well) onto the previously prepared under layers.

## Xenograft tumor model

All experiments involving animals were performed with approval from the Virginia Commonwealth University Institutional Animal Care and Use Committee (IACUC; protocol #: AD20282). Control RWPE-1 or PLK1-overexpressing cells ($1 \times 10^6$) were mixed with an equal volume of BD Matrigel Basement Membrane Matrix and injected subcutaneously into the flanks of NOD/SCID/$\gamma_c^{null}$ mice (6-week-old male mice, from The Jackson Laboratory, stock no. 005557). Seven mice were used per group. Mice were housed under specific-pathogen-free conditions. Tumor growth was observed for 5 weeks. Tumors were measured 3 times weekly with a caliper and their volumes calculated using

the following formula: π × [length in millimeters] × [width in millimeters]/6. At the end of the experiment, mice were euthanized and the tumors and lungs were removed and preserved for pathological examination. Formalin-fixed sections were stained with hematoxylin and eosin. Immunochemistry for PSA on paraffin-embedded tissue sections (5–6 μm) was done using a catalyzed amplification system (Dako, Carpinteria, CA) as we described previously (*Fu et al., 2009*).

## Statistical analysis

The statistical tests used are indicated in the figure legends. All statistical tests were performed using GraphPad Prism version 6.02 for Windows (GraphPad Software). Data are expressed as mean ± s.e.m., and $p < 0.05$ was considered statistically significant. A two-tailed Student's *t*-test was used to compare differences between treated groups and their paired controls. To compare the track distance and velocity between 2 groups, a two-tailed Mann-Whitney rank sum test was performed. Groups of 7 mice were used for the tumor xenograft experiments. No statistical method was used to pre-determine sample size. The experiments were not randomized. The investigators were blinded to allocation during the xenograft tumor model experiments. For other experiments, the investigators were not blinded to allocation during experiments and outcome assessment.

## Acknowledgements

This work was supported by grants from the American Cancer Society (ACS Research Scholar Grant 127626-RSG-15-005-01-CCG to ZF, and an institutional grant to ZF), the National Institutes of Health (NIH R01 CA191002 to ZF), National Institutes of Health (P50 CA058236 to Martin G Pomper and PBF), and the National Cancer Institute (NCI Cancer Center Support Grant P30CA016059 to VCU Massey Cancer Center). The authors thank Sarah Conine for editorial assistance with the manuscript. PBF holds the Thelma Newmeyer Corman Chair in Cancer Research in the VCU Massey Cancer Center.

## Additional information

### Competing interests

PBF: Co-founder of serves as a consultant to and has ownership interest in CTS, Inc. Johns Hopkins University, Virginia Commonwealth University has ownership interest in CTS, Inc. Competing financial interests had no role in study design, data collection and interpretation, or the decision to submit the work for publication. The other authors declare that no competing interests exist.

### Funding

| Funder | Grant reference number | Author |
| --- | --- | --- |
| American Cancer Society | Research Scholar Grant, 127626-RSG-15-005-01-CCG | Zheng Fu |
| National Institutes of Health | R01 CA191002 | Zheng Fu |
| National Institutes of Health | P50 CA058236 | Paul B Fisher |

The funders had no role in study design, data collection and interpretation, or the decision to submit the work for publication.

### Author contributions

JW, Acquisition of data, Analysis and interpretation of data; AII, PBF, Acquisition of data, Drafting or revising the article; ZF, Conception and design, Acquisition of data, Analysis and interpretation of data, Drafting or revising the article

### Author ORCIDs

Zheng Fu, http://orcid.org/0000-0003-0837-3728

## Ethics

Animal experimentation: Animal experimentation: This study was performed in strict accordance with the recommendations in the Guide for the Care and Use of Laboratory Animals of the National Institutes of Health. All of the animals were handled according to approved institutional animal care and use committee (IACUC; protocol #: AD20282) protocols of the Virginia Commonwealth University.

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
