## [Decision Letter]

Thank you for submitting your work entitled "PLK1 induces epithelial-to-mesenchymal transition and promotes epithelial cell motility by activating CRAF/ERK signaling" for peer review at *eLife*. Your submission has been favorably evaluated by Tony Hunter (Senior Editor) and two reviewers.

The reviewers have discussed the reviews with one another and the editor has drafted this decision to help you prepare a revised submission.

These studies show that overexpression of Polo-like kinase 1 (PLK1) was able to induce EMT in prostate epithelial cells. Mechanistically, overexpression of PLK1 was found to activate MEK/ERK via direct phosphorylation and stabilization of CRAF at S338. Activated ERK kinase has been reported by others to upregulate FRA1, which in turn induces ZEB1/2 to promote EMT, consistent with their conclusions. The reviewers' main concern is that essentially all the experiments rely on PLK1 overexpression, which may or may not be physiological, with only two PLK1 knockdown experiments shown in Figure 3 and Figure 8.

Essential revisions:

1) Throughout the study, there is heavy reliance on overexpression of PLK1 in prostate epithelial cells. This raises the question how much Flag-PLK1 was overexpressed compared to endogenous PLK1 (what fraction of cells were productively infected in Figure 1?), and how this relates to the level of PLK1 expression in tumor cells known to overexpress PLK1, and expression levels need to be documented. More importantly, although overexpression of PLK1 was able to induce EMT, the data in Figure 3 about endogenous PLK1 on EMT are not convincing. First, only a single shRNA was used and there was no re-expression control (see concerns in point 3). Second, it is unclear whether PLK1 is differentially expressed and/or activated in the metastatic cell lines used in this figure compared to the non-metastatic lines used in the rest of the study. In this regard, it would also be important to determine, whether PLK1 is involved in any physiological EMT events, such as TGFβ-induced EMT.

2) Although the effect of endogenous PLK1 in cell motility is well demonstrated, its role in EMT is still in question. Immunoblotting analyses of four EMT gene products are not sufficient evidence to support the conclusion that knockdown of PLK1 blocked EMT and caused partial MET. Morphological and immunostaining analyses are needed in addition. In addition, a standard system to analyze EMT in prostate cancer is "ARCaP, ARCaP-E and ARCaP-M", and these gold standard cell lines for studying EMT in prostate cancer field could be used to verify the EMT results.

3) In Figure 8 a single shRNA was used to knock down PLK1, and only CRAF phosphorylation was monitored. This experiment needs to be repeated with additional PLK1 shRNAs, combined with re-expression of shRNA-resistant WT and kinase-dead PLK1 to establish that the observed effect on CRAF effect is due to loss of PLK1 kinase activity. This is particularly important because the level of CRAF protein was significantly reduced by the PLK1 shRNA, which could account at least in part for its decreased phosphorylation. At the same time, the consequences of knocking down endogenous PLK1 on activation of the rest of the ERK/FRA1/ZEB pathway needs to be analyzed using the same controls. Since PLK1 is generally believed to be essential for cell cycle progression, the authors should also analyze the cell cycle distribution of the cells to determine if this might account for any observed change when PLK1 is knocked down. In addition, depletion of Plk1 not only enriches cells in mitosis, but subsequently induces apoptosis in a broad range of cancer cells, and this could also attenuate migration and lead to a reduced expression or decay/degradation of mesenchymal markers (Figure 3) and CRAF (Figure 8). For this reason, it would also be informative to carry out and include apoptosis assays.

4) The authors need to rule out that PLK1 overexpression in the epithelial cell lines does not induce autocrine stimulation of the CRAF pathway, which would also increase pS338 levels.

5) The authors need to acknowledge and discuss the relevance prior work in this area to their own findings. For example, Mielgo et al. (Nat Med. 17:1641) reported that PLK1 and CRAF associate, but they concluded that CRAF pS338 is upstream of PLK1 activation, rather than, as you propose, PLK1 being upstream of CRAF activation. In addition, Jiang and Tang (Respir Res 16:93) reported that PLK1 can activate MEK/ERK, although that study suggested the activation was CRAF pS338-independent, with PLK1 phosphorylating MEK activating sites. Finally, much of the data in Figure 4, Figure 5 and Figure 6 replicates what Shin et al. (Mol Cell 38:114) reported for v-Ras transformed MCF-10A cells undergoing EMT.

These points need to be addressed in a revised version.

---

## [Author Response]

*Essential revisions: 1) Throughout the study, there is heavy reliance on overexpression of PLK1 in prostate epithelial cells. This raises the question how much Flag-PLK1 was overexpressed compared to endogenous PLK1 (what fraction of cells were productively infected in Figure 1?), and how this relates to the level of PLK1 expression in tumor cells known to overexpress PLK1, and expression levels need to be documented.*

The RWPE-1–PLK1 cells are a stable cell line established from a mixed population of multiple clones to avoid clonal variation. One hundred percent of RWPE-1 cells express flag-tagged PLK1, as determined by immunofluorescence labeling with anti-Flag antibodies (data not shown). As shown in Figure 1 of the revised manuscript, RWPE-1–PLK1 cells express PLK1 at a level comparable to metastatic C4-2B PCa cells, which is also comparable to other metastatic PCa cell lines (Figure 1).

*More importantly, although overexpression of PLK1 was able to induce EMT, the data in Figure 3 about endogenous PLK1 on EMT are not convincing. First, only a single shRNA was used and there was no re-expression control (see concerns in point 3). Second, it is unclear whether PLK1 is differentially expressed and/or activated in the metastatic cell lines used in this figure compared to the non-metastatic lines used in the rest of the study.*

We appreciate this comment and performed additional experiments to demonstrate the role of endogenous PLK1 on EMT. We have included 2 different shRNAs targeting the 3’-UTR of PLK1 and re-expression of PLK1 in PLK1-knockdown cells. The new results are now presented in Figure 3 and the results in the previous version of Figure 3 have been moved to Figure 3—figure supplement 2. Furthermore, we compared PLK1 expression and activity in a panel of PCA cell lines and found that PLK1 is differentially expressed and/or activated in PCa cell lines (higher in the metastatic PCa cell lines and lower in the non-metastatic cell lines) (Figure 1). Furthermore, the expression level and activity of PLK1 are much higher in highly metastatic ARCaP_M_ cells than in less metastatic ARCaP_E_ cells (Figure 3—figure supplement 3), further suggesting a role for PLK1 in PCa metastasis.

*In this regard, it would also be important to determine, whether PLK1 is involved in any physiological EMT events, such as TGFβ-induced EMT.*

We appreciate the reviewers’ suggestion. We have performed the recommended experiment and present the corresponding data in Figure 3—figure supplement 4 of the revised manuscript. We observed that PLK1 inhibition attenuated EMT induced by a co-treatment of RWPE and ARCaP_E_ cells with TGFβ and EGF (Figure 3—figure supplement 4).

*2) Although the effect of endogenous PLK1 in cell motility is well demonstrated, its role in EMT is still in question. Immunoblotting analyses of four EMT gene products are not sufficient evidence to support the conclusion that knockdown of PLK1 blocked EMT and caused partial MET. Morphological and immunostaining analyses are needed in addition. In addition, a standard system to analyze EMT in prostate cancer is "ARCaP, ARCaP-E and ARCaP-M", and these gold standard cell lines for studying EMT in prostate cancer field could be used to verify the EMT results.*

As the reviewers suggested, we have performed additional experiments (morphological and immunostaining analyses) to more comprehensively demonstrate the role of endogenous PLK1 in EMT (Figure 3, Figure 3—figure supplement 2). Furthermore, we validated the role of PLK1 in EMT using the ARCaP model (ARCaP-E and ARCaP-M cells) (Figure 3—figure supplement 3). These studies were performed in collaboration with Dr. Paul Fisher, a co-author on this paper, who obtained these cells from Novicure Biotechnology under an MTA agreement. This agreement specifically prevents providing these cells to other investigators or developing stable variants of these cells. Based on these limitations, we employed transient transfection assays to upregulate and downregulate PLK1 in ARCaP_E_ and ARCaP_M_ cells, respectively, to address the specific comments of the reviewers. The results obtained with several gold standard cell lines (presented in Figure 3 and Figure 3—figure supplement 1–Figure 3—figure supplement 3) support our conclusion about PLK1-dependent regulation of EMT in prostate cancer cells.

*3) In Figure 8 a single shRNA was used to knock down PLK1, and only CRAF phosphorylation was monitored. This experiment needs to be repeated with additional PLK1 shRNAs, combined with re-expression of shRNA-resistant WT and kinase-dead PLK1 to establish that the observed effect on CRAF effect is due to loss of PLK1 kinase activity. This is particularly important because the level of CRAF protein was significantly reduced by the PLK1 shRNA, which could account at least in part for its decreased phosphorylation. At the same time, the consequences of knocking down endogenous PLK1 on activation of the rest of the ERK/FRA1/ZEB pathway needs to be analyzed using the same controls.*

We appreciate this suggestion and have performed the recommended experiments, which include 2 different shRNAs along with re-expression of WT and kinase-dead PLK1. In addition to the effect on CRAF phosphorylation, the consequences of knocking down endogenous PLK1 on activation of the rest of the ERK/FRA1/ZEB pathway were monitored. As shown in Figure 8 and Figure 8—figure supplement 1 of the revised manuscript, loss of PLK1 kinase activity led to reduction in CRAF phosphorylation at S338 and S339, and downstream ERK/FRA1/ZEB1/2 signaling.

*Since PLK1 is generally believed to be essential for cell cycle progression, the authors should also analyze the cell cycle distribution of the cells to determine if this might account for any observed change when PLK1 is knocked down. In addition, depletion of Plk1 not only enriches cells in mitosis, but subsequently induces apoptosis in a broad range of cancer cells, and this could also attenuate migration and lead to a reduced expression or decay/degradation of mesenchymal markers (Figure 3) and CRAF (Figure 8). For this reason, it would also be informative to carry out and include apoptosis assays.* We appreciate the reviewers’ suggestion and have examined the cell cycle distribution and apoptosis in those cells. It is noteworthy that in order to avoid the effects of severe PLK1 depletion on growth inhibition and cell death, we used partial depletion of endogenous PLK1 in this study. Partial PLK1 knockdown did not significantly affect cell cycle progression or induce apoptosis (Figure 3—figure supplement 1).

*4) The authors need to rule out that PLK1 overexpression in the epithelial cell lines does not induce autocrine stimulation of the CRAF pathway, which would also increase pS338 levels.*

As the reviewers suggested, we have performed an additional experiment to rule out the possibility that PLK1-induced activation of the CRAF pathway is due to autocrine stimulation. We transiently inhibited PLK1 activity in mitotic shake-off cells by treating cells with BI 2536, a pharmacological PLK1 inhibitor, (at a low and a high concentration) for a short period of time (30 min). Phosphorylation of CRAF at S338 and S339 was dramatically decreased immediately upon brief exposure to the PLK1 inhibitor (Figure 8), which provides another line of evidence that PLK1 directly phosphorylates CRAF at S338 and S339.

*5) The authors need to acknowledge and discuss the relevance prior work in this area to their own findings.*

We appreciate this suggestion and have included the acknowledgement and discussion of the relevant prior work in this area to our findings in the Discussion section.

*For example, Mielgo et al. (Nat Med. 17:1641) reported that PLK1 and CRAF associate, but they concluded that CRAF pS338 is upstream of PLK1 activation, rather than, as you propose, PLK1 being upstream of CRAF activation.*

We have included additional evidence in Figure 8 demonstrating that PLK1 directly phosphorylates CRAF. Mielgo et al. reported that PLK1 associates with CRAF and this interaction does not require CRAF kinase activity. Interestingly, they also found that CRAF indirectly promotes PLK1 activation (Nat Med. 17:1641). This would add an additional layer of complexity to the PLK1-CRAF interplay (as we proposed) to further activate CRAF.

*In addition, Jiang and Tang (Respir Res 16:93) reported that PLK1 can activate MEK/ERK, although that study suggested the activation was CRAF pS338-independent, with PLK1 phosphorylating MEK activating sites.*

Interestingly, Jiang and Tang reported that PLK1 can directly activate MEK/ERK signaling through phosphorylation of a MEK activating site in airway smooth muscle cells (Respir Res 16:93); however, that is not the case for prostate cancer (Figure 8). This would suggest that PLK1 may differentially trigger distinct signaling pathways under different physiological conditions.

*Finally, much of the data in Figure 4, Figure 5 and Figure 6 replicates what Shin et al. (Mol Cell 38:114) reported for v-Ras transformed MCF-10A cells undergoing EMT.*

It has been reported that ERK2/Fra1/ZEB1/2 signaling is responsible for EMT induction in RasG12V transformed MCF10A cells (*Mol Cell* 38:114). Our results convincingly demonstrate that PLK1 functions as a key regulator of EMT and cell motility in normal prostate epithelium and prostate cancer. The signaling mechanisms underlying the observed cellular effects of PLK1 involve direct PLK1-dependent phosphorylation of CRAF with subsequent stimulation of the MEK1/2-ERK1/2-Fra1-ZEB1/2 signaling pathway. Our new findings along with the results from this other group suggest that ERK/Fra1/ZEB signaling could be a common pathway to induce EMT in mammalian epithelial cells.